# ECHOSAT: Estimating Canopy Height Over Space And Time

## Abstract

Forest monitoring is critical for climate change mitigation. However, existing global tree height maps provide only static snapshots and do not capture temporal forest dynamics, which are essential for accurate carbon accounting. We introduce ECHOSAT, a global and temporally consistent tree height map at $10\,\mathrm{m}$ resolution spanning multiple years. To this end, we resort to multi-sensor satellite data to train a specialized vision transformer model, which performs pixel-level temporal regression. A self-supervised growth loss regularizes the predictions to follow growth curves that are in line with natural tree development, including gradual height increases over time, but also abrupt declines due to forest loss events such as fires. Our experimental evaluation shows that our model improves state-of-the-art accuracies in the context of single-year predictions. We also provide the first global-scale height map that accurately quantifies tree growth and disturbances over time. We expect ECHOSAT to advance global efforts in carbon monitoring and disturbance assessment. The produced height maps will be made accessible upon acceptance.

## 1 Introduction

Forests play a crucial role in the mitigation of climate change, absorbing $3.5\,\mathrm{Pg}$ of carbon per year, which represents almost half of anthropogenic fossil fuel emissions (Pan et al., 2024). As global carbon emissions continue to increase, precise monitoring of forest carbon dynamics using up-to-date information on forest health and carbon balance has become an essential for effective climate policy and forest management decisions. Recent advances in satellite remote sensing and machine learning have enabled automated forest carbon monitoring on country-to-global scales, using tree height as a key proxy for estimating the so-called above-ground biomass (AGB) and, therefore, carbon storage (Schwartz et al., 2023). Most of these height maps provide a static representation of forests at a specific point in time and cannot be used to estimate year-to-year carbon absorption (Tolan et al., 2024; Pauls et al., 2024; Lang et al., 2023; Potapov et al., 2021).

While such static snapshots of forests worldwide already depict a viable resource, they do not capture temporal dynamics such as tree growth or forest loss. Some studies provide such a temporal monitoring of forests. However, they are often limited to large scale disturbances such as forest losses due to big fires (Reiche et al., 2021; Hansen et al., 2013b). Small-scale height decreases from degradation, individual tree mortality or forest thinning, however, are significantly smaller and, hence, harder to detect. Additionally, very few studies have succeeded in retrieving realistic year-to-year forest growth pattern at a high resolution (Turubanova et al., 2023; Schwartz et al., 2025), and rely on single-year models independently applied to multiple years along with extensive post-processing to achieve temporal consistency. None of the aforementioned approaches is based on models that inherently learn forest temporal dynamics, thus, when no post-processing is applied, this leads to unrealistic fluctuations at the pixel-level and poor temporal coherence in predictions.

In this work, we provide the first global tree height mapping approach at high resolution ($10\,\mathrm{m}$) across multiple years. Our method combines a transformer-based temporal regression model with an adapted loss that addresses sparse temporal supervision, where labels are limited both spatially (not every pixel has ground truth) and temporally (each pixel often has only a single measurement), while enforcing physically realistic growth patterns. By leveraging multi-sensor satellite data, we produce a coherent global time series of tree height maps at unprecedented scale and resolution.

**Contributions.** Our main contributions are threefold. First, we present ECHOSAT, the first high-resolution (10 m) spatio-temporal tree height map covering the entire globe across seven years (2018–2024), which enables reliable monitoring of forest dynamics and disturbances at scale. Second, to enforce physically realistic forest growth patterns, we introduce a novel growth loss framework specifically designed for training temporal regression models with sparsely distributed and temporally irregular ground truth labels. Third, we demonstrate that our model inherently learns realistic temporal forest height dynamics without relying on post-processing, capturing both natural growth and abrupt disturbances. We further demonstrate that our model outperforms existing approaches on single-year evaluations.

## 2 BACKGROUND

We construct a consistent global time series of forest heights from multiple satellite datasets. Remote sensing has long been used to complement and upscale forest inventory measurements (Tomppo et al., 2008), and more recently deep learning approaches have been introduced in this context. We briefly review these methods and highlight the relevance and impact of our work in this context.

### 2.1 VISION ARCHITECTURES AND SELF-SUPERVISED LEARNING

Recent advances in remote sensing have been driven by foundation models adapted from computer vision (Tseng et al., 2025; Fuller et al., 2023; Fayad et al., 2025; Astruc et al., 2025; Tseng et al., 2023). These models predominantly employ isotropic architectures (e.g., standard Vision Transformers) to leverage scalable self-supervised learning (SSL) objectives. The three dominant paradigms are: (1) Masked Image Modeling (MIM), such as MAE (He et al., 2022), which learns by reconstructing randomly masked patches; (2) Contrastive Learning, such as SimCLR(Chen et al., 2020), which optimizes for semantic invariance between different views or crops of an image; or (3) Joint-Embedding Prediction Architectures like I-JEPA (Assran et al., 2023) that predict the embedding of one part of the image using another part of the same image.

However, these paradigms present structural incompatibilities with hierarchical architectures (Liu et al., 2021; Cao et al., 2022), which are otherwise superior for dense prediction tasks. Specifically, the unstructured masking strategies central to MAE and I-JEPA disrupt the rigid grid alignment required by the shifted-window attention mechanisms in hierarchical transformers. Furthermore, contrastive objectives often enforce global semantic uniformity, suppressing the high-frequency, pixel-level spatial details required for fine-grained regression. Consequently, while Swin-based architectures offer inductive biases well-suited for dense canopy height estimation, they have seen limited adoption in large-scale remote sensing foundation models.

### 2.2 FOREST HEIGHT PREDICTION USING REMOTE SENSING DATA

Satellite remote sensing at high resolution employs mainly three types of sensors: optical, SAR (Synthetic Aperture Radar) and LiDAR (Light Detection And Ranging). Optical sensors operate passively, measuring sun's reflected electromagnetic radiation across multiple spectral bands from visible to near-infrared wavelengths. For instance Sentinel-2 delivers multi-spectral optical imagery with up to 10 m spatial resolution and approximately 6-day revisit time depending on latitude, while Landsat provides historical multi-spectral data with 30 m spatial resolution, enabling long-term temporal analysis. In contrast, SAR sensors actively transmit microwave signals and measure the backscattered energy, enabling data acquisition regardless of illumination conditions and cloud cover. LiDARs are light-emitting and receiving sensors that estimate distances by measuring the time it takes for the light to return to the sensor after being reflected on an object. The Global Ecosystem Dynamics Investigation (GEDI) mission, operated by the NASA and deployed on the International Space Station (ISS), provides spaceborne LiDAR measurements of forest vertical structure within 25 m diameter footprints end of 2018 (Dubayah et al., 2022). This data can be used to get (aboveground) height measurements of the footprint. GPS and star tracker data are used to estimate the position of ISS and deduce geolocation of a measurement.

Due to it's correlation with forest biomass, forest height mapping has gained significant attention in recent years, with numerous studies producing tree height maps at regional (Favrichon et al., 2025), national (Su et al., 2025; Schwartz et al., 2023), continental (Liu et al., 2023) and global

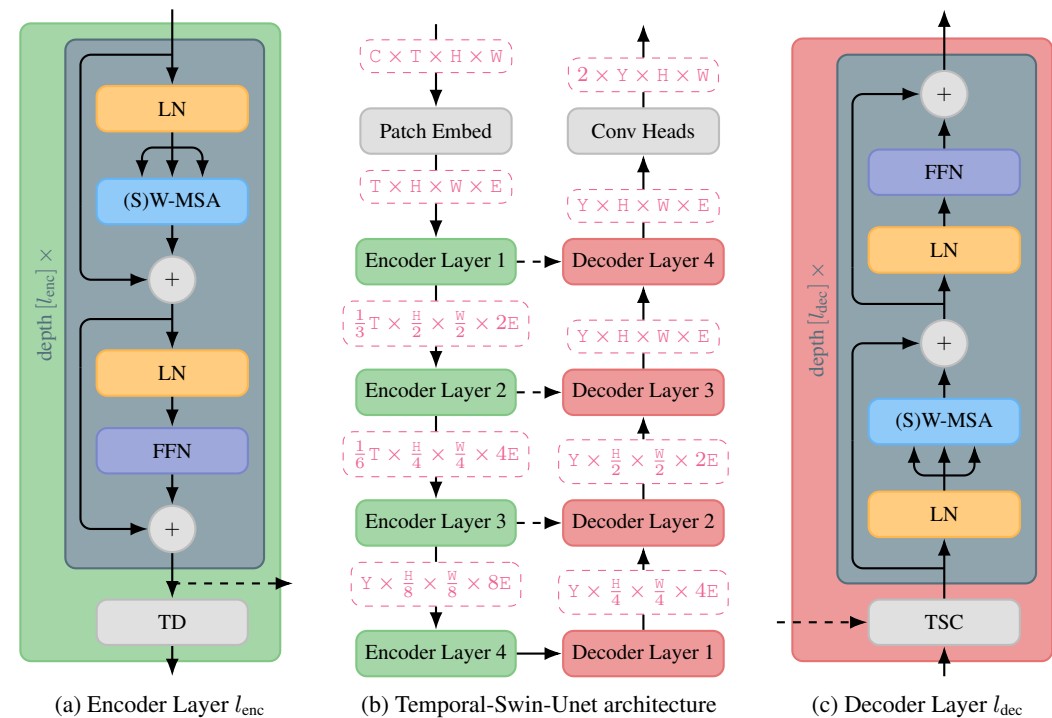

(a) Encoder Layer $l_{enc}$      (b) Temporal-Swin-Unet architecture      (c) Decoder Layer $l_{dec}$

Figure 1: Architecture of the Temporal-Swin-Unet. Skip connections are depicted as dashed arrows. The shape of the tensor in between a layer is shown in magenta: $C$ (channels), $T$ (timesteps), $H$ (height), $W$ (width), $Y$ (years), and $E$ (embedding dimension). In addition to the Video Swin Transformer Blocks (Liu et al., 2022), Encoder Layers have a *Temporal Downsample* (TD) layer at the end, and Decoder Layers a *Temporal Skip Connection* (TSC) at the beginning.

scale. These maps typically combine remote sensing imagery with reference height measurements from spaceborne LiDAR systems such as GEDI or ICESat, or from airborne laser scanning (ALS) campaigns. The development of global tree height maps has progressed significantly in recent years. Potapov et al. (2021) pioneered the first global tree height map using Landsat data at 30 m resolution, GEDI measurements, and a random forest model. Subsequent work by Lang et al. (2023) improved spatial resolution to 10 m using Sentinel-2 data and convolutional neural networks. More recently, Pauls et al. (2024) developed a global map using a UNet architecture with a specialized loss function designed to improve robustness to noise, while Tolan et al. (2024) achieved individual tree-level detection using a DINOv2 model fine-tuned on 1 m+ Maxar data with ALS and GEDI labels.

Single-snapshot estimates cannot capture the effects of management or climate change over time. Temporal tree height mapping methods address this, with the simplest approach training single-year models independently on each year of remote sensing data (Kacic et al., 2023). A more sophisticated approach employs space-for-time substitution, where models trained on spatial variations are applied to temporal sequences under the assumption that similar spatial patterns correspond to similar temporal dynamics (Schwartz et al., 2025). A third approach uses classical machine learning methods with extensive post-processing to smooth temporal inconsistencies and reduce prediction uncertainty (Turubanova et al., 2023). These approaches easily capture abrupt large-scale height changes due to forest clearcuts or large disturbance events (fires, storms) but often overlook small disturbances at the tree level. Above all, they cannot produce consistent height time-series at the pixel level which preclude any detailed carbon dynamics analysis.

## 2.3 RELEVANCE AND IMPACT

Accurate mapping of tree height is a prerequisite for assessing biomass carbon storage and wood resources over the world forests. For monitoring forest changes under human and climate pressure, we need dynamic maps instead of static products. Furthermore, as most height loss instances come from

small scale events such as mortality occurring in clusters of trees, natural forest disturbances, and human activities including timber harvest, degradation and deforestation, a high spatial resolution is needed to capture the fine scale patterns of height decreases.

The new global annual forest height maps at $10\,\text{m}$ resolution developed in this study with a deep learning model that can learn to reconstruct height changes not only from spatial gradients but also from temporal images represent a significant step forward that goes beyond previous global static maps and extends temporal change maps that were limited to few regions and used coarser resolution models. Our map was evaluated against height labels not used for training, but coming from the same space-borne LiDAR. Over pixels not affected by losses where forests are growing or regrowing after previous loss events, we also showed regular year on year increment of height that are consistent with ecological knowledge indicating that younger and shorter trees grow faster than taller ones.

The main remaining challenge is the verification of our predicted height changes against independent observations such as airborne LiDAR repeated campaigns, dense ground-based inventories census, which include revisits of hundreds of forest plots over time, and interpretation of high resolution imagery for height loss events. Current approaches to temporal tree height mapping rely on post-processing techniques to achieve temporal consistency, as existing models are not inherently designed to learn realistic temporal dynamics. This represents a significant limitation, as models that could naturally incorporate temporal constraints and learn realistic growth patterns would provide more accurate and physically meaningful predictions without requiring extensive smoothing or correction procedures.

## 3 APPROACH

With the research gap in mind, we develop a new methodology to estimate tree height with coherent temporal predictions at global scale by training the model inherently to produce realistic temporal changes. The approach uses a model with two outputs heads and a two-step process: the first (reference) head is pre-trained using Huber loss and the second (prediction) head is finetuned on pseudo-labels created from the frozen first head. Further details on data processing, quality filtering, normalization, and the model architecture are provided in Appendix A.2.

### 3.1 DATA

We integrate multi-temporal satellite data spanning 2018–2024 to enable global-scale temporal tree height mapping, with Sentinel-2 providing the primary image source at $10\,\text{m}$ resolution.

**Multi-sensor Satellite Data.** We combine optical (Sentinel-2) and radar backscatter (Sentinel-1, ALOS PALSAR-2) data with auxiliary products (TanDEM-X DEM and forest classification). Sentinel-2 provides monthly images at $10\,\text{m}$ resolution across 12 spectral bands, while radar data offers quarterly (Sentinel-1) and yearly (ALOS PALSAR-2) composites. GEDI LiDAR measurements serve as ground truth labels for 2019–2024 with approximately $25\,\text{m}$ diameter footprints.

**Data Processing Pipeline.** We create a unified input tensor of shape $18 \times 84 \times 96 \times 96$ (channels $\times$ timesteps $\times$ height $\times$ width) by temporally aligning different input data and spatially resampling all data to $10\,\text{m}$ resolution. Quality filtering on GEDI ground truth ensures reliable measurements.

### 3.2 MODEL ARCHITECTURE

Our model is based on the Swin Transformer (Liu et al., 2021) and leverages two key extensions: the Video Swin Transformer (Liu et al., 2022), designed for video input processing, and the Swin-Unet (Cao et al., 2022), tailored for semantic segmentation tasks.

**Temporal-Swin-Unet.** We combine the extensions from Cao et al. (2022) and Liu et al. (2022) with some small, but crucial, changes to perform pixel-wise regression on a time-series of images of shape $C \times T \times H \times W$. We call the resulting architecture Temporal-Swin-Unet, depicted in Figure 1. Different from most contemporary approaches, we adopt a patch size of $1 \times 1$ pixels, following Nguyen et al. (2025). The Patch Embed layer linearly projects each voxel[1] into the embedding

---

[1] We define a voxel as a value in the 3D grid $T \times H \times W$, i.e. a pixel at a given timestep.

dimension $E$. Operating on the original resolution is crucial for our application, where every pixel corresponds to $10 \times 10$ meters. The model consists of four Encoder and Decoder layers each, which are connected via skip connections. Each Encoder and Decoder layer consists of multiple Video Swin Transformer Blocks (Liu et al., 2022). Except at the Unet's lowest level, all Encoder layers end with a *Temporal Downsample* (TD) layer and all Decoder layers begin with a *Temporal Skip Connection* (TSC).

**TD and TSC layer.** The TD layer reduces the temporal and spatial dimension by applying a year-wise linear projection, concatenating the embeddings of four adjacent pixels, and performing another linear projection to double the embedding size. The TSC layer enriches the Decoder-features token-wise per year with the corresponding Encoder-features of the same year via a Transformer layer. At the end of the Decoder layer, we perform spatial upsampling to increase the spatial resolution by a factor of two.

**Conv Heads.** On top of the final Decoder layer we use two heads: the reference head, which is used for pretraining and projects the embeddings voxel-wise to scalar values; the prediction head is added later for fine-tuning and consists of three Conv3D layers with normalization and activation layers in between. Our model thus outputs a tensor of shape $2 \times Y \times H \times W$, being two canopy height predictions per year and pixel.

### 3.3 GROWTH LOSS

**Motivation and Notation.** We propose a self-supervised approach to achieve consistent growth curves, which are monotonically increasing, but allow for sharp cut-offs in disturbance situations. Let $\mathbf{Y}^{\mathrm{ref}} \in \mathbb{R}^{Y \times H \times W}$ and $\mathbf{Y}^{\mathrm{pred}} \in \mathbb{R}^{Y \times H \times W}$ be the outputs of the reference head and the prediction head. Furthermore, let $\boldsymbol{z}^{\mathrm{ref}} := \mathbf{Y}^{\mathrm{ref}}_{:,h,w} \in \mathbb{R}^Y$ and $\boldsymbol{z}^{\mathrm{pred}} := \mathbf{Y}^{\mathrm{ref}}_{:,h,w} \in \mathbb{R}^Y$ be the predicted time series at the pixel $(h, w) \in \{1, \ldots, H\} \times \{1, \ldots, W\}$. In short, the loss works as follows: it fits a regression on $\boldsymbol{z}^{\mathrm{ref}}$ and uses the fitted values as pseudo-labels for $\boldsymbol{z}^{\mathrm{pred}}$. The regression function is either linear or a combination of two linear functions (pre- and post-disturbance) in case a disturbance is detected, where all slopes are forced to lie in a reasonable interval for tree growth, e.g. in $[s_{\min} = 0\,\mathrm{m/year}, s_{\max} = 3\,\mathrm{m/year}]$.

**Disturbance Indicator.** A disturbance is considered to occur in $\boldsymbol{z}^{\mathrm{ref}} \in \mathbb{R}^Y$ when a) tree height decreased by more than $50\%$ and more than $4\,\mathrm{m}$ and b) tree height decreased to less than $10\,\mathrm{m}$ within two years.[2] Thus, we define the set of pre-disturbance years $\mathbb{Y}_{\mathrm{dstb}}(\boldsymbol{z}^{\mathrm{ref}})$ to be

$$\mathbb{Y}_{\mathrm{dstb}}(\boldsymbol{z}^{\mathrm{ref}}) = \{y \in \{1, \ldots, Y-1\} \mid \boldsymbol{z}^{\mathrm{ref}}_{y+1} \leq \min(0.5 \cdot \boldsymbol{z}^{\mathrm{ref}}_y, \boldsymbol{z}^{\mathrm{ref}}_y - 4), \min(\boldsymbol{z}^{\mathrm{ref}}_{y+1}, \boldsymbol{z}^{\mathrm{ref}}_{y+2}) \leq 10\}.$$

The local disturbance indicator, defined as the final year preceding a disturbance, is defined by

$$\mathbb{I}_{\mathrm{dstb,loc}}(\boldsymbol{z}^{\mathrm{ref}}) := \begin{cases} Y & \text{if } \mathbb{Y}_{\mathrm{dstb}}(\boldsymbol{z}^{\mathrm{ref}}) = \emptyset \\ \min(\mathbb{Y}_{\mathrm{dstb}}(\boldsymbol{z}^{\mathrm{ref}})) & \text{else} \end{cases} \in \{1, \ldots, Y\}.[3]$$

Combining the pixel-wise local disturbance indicator, we can build an image-wise local disturbance indicator $\mathbb{I}_{\mathrm{dstb,loc}}(\mathbf{Y}^{\mathrm{ref}}) \in \{1, \ldots, Y\}^{H \times W}$. The disturbance indicator is finally defined as

$$\mathbb{I}_{\mathrm{dstb}}(\mathbf{Y}^{\mathrm{ref}}) = \mathrm{MinPool}_{3 \times 3}(\mathbb{I}_{\mathrm{dstb,loc}}(\mathbf{Y}^{\mathrm{ref}})) \in \{1, \ldots, Y\}^{H \times W}. \tag{1}$$

**Constrained Linear Regression.** For some $N \in \mathbb{N}$ and vector $\boldsymbol{z} \in \mathbb{R}^N$, we define the constrained linear regression vector $\hat{\boldsymbol{z}} \in \mathbb{R}^N$ with respect to a minimal and maximal slope $s_{\min} < s_{\max}$ as follows. Let $\tilde{s} \in \mathbb{R}$ be the slope of the simple linear regression model for the dataset $\{(1, \boldsymbol{z}_1), (2, \boldsymbol{z}_2), \ldots, (N, \boldsymbol{z}_N)\}$. Then the slope $s$, the intercept $b$ and $\hat{\boldsymbol{z}}$ are defined by

$$s := \min(\max(\tilde{s}, s_{\min}), s_{\max}) \in [s_{\min}, s_{\max}]$$

$$b := \bar{z} - s \cdot \frac{N+1}{2} \in \mathbb{R} \text{ with } \bar{z} := \frac{1}{N} \sum_{n=1}^N \boldsymbol{z}_n$$

$$\hat{\boldsymbol{z}} := s \cdot [1, 2, \ldots, N]^{\mathrm{T}} + b \in R^N.$$

---

[2]As no global temporal tree height dataset is available, validation of these set values is difficult, hence we solely rely on domain expert knowledge.

[3]Please note that for the majority of tree height prediction time series, there is at most one disturbance year and the minimum is just taken to take care of the other rare cases.

Table 1: Comparison of global-scale methods for 2020 regarding MAE (m), MSE (m$^2$), RMSE (m), MAPE (%), $R^2$ and $R^2_{\mathrm{all}}$ (on all labels, including labels below $5\,\mathrm{m}$). IQR is given in square brackets.

| Method | MAE ↓ | MSE ↓ | RMSE ↓ | MAPE ↓ | $R^2$ ↑ | $R^2_{\mathrm{all}}$ ↑ |
|---|---|---|---|---|---|---|
| Potapov et al. (2021) | 9.11 [7.73] | 185.52 [107.99] | 13.62 [7.73] | 54.90 [78.99] | 0.50 | 0.70 |
| Lang et al. (2023) | 7.97 [6.67] | 143.78 [81.04] | 11.99 [6.67] | 53.33 [76.84] | 0.52 | 0.71 |
| Pauls et al. (2024) | 6.85 [6.41] | 138.19 [60.75] | 11.76 [6.41] | 34.20 [33.16] | 0.51 | 0.73 |
| Tolan et al. (2024) | 11.89 [9.09] | 260.28 [182.25] | 16.13 [9.09] | 68.78 [57.80] | 0.45 | 0.64 |
| Ours | **5.85** [4.90] | **118.07** [36.16] | **10.87** [4.90] | **30.20** [33.32] | **0.59** | **0.77** |

**Growth loss.** Pseudo-labels are created by performing piecewise constrained linear regression on the reference time series. Let $y := \mathbb{I}_{\mathrm{dstb}}(\mathbf{Y}^{\mathrm{ref}})_{h,w} \in \{1, 2, \ldots, Y\}$ be the detected pre-disturbance year of the reference output and split $\boldsymbol{z}^{\mathrm{ref}}$ into pre- and post-disturbance vectors, that is

$$\boldsymbol{z}^{\mathrm{ref}}_{\mathrm{pre}} := [\boldsymbol{z}^{\mathrm{ref}}_1, \ldots, \boldsymbol{z}^{\mathrm{ref}}_y]^{\mathrm{T}} \in \mathbb{R}^y \text{ and } \boldsymbol{z}^{\mathrm{ref}}_{\mathrm{post}} := [\boldsymbol{z}^{\mathrm{ref}}_{y+1}, \ldots, \boldsymbol{z}^{\mathrm{ref}}_Y]^{\mathrm{T}} \in \mathbb{R}^{Y-y}.$$

Then the pseudo-labels are defined by concatenating the constrained linear regression vectors $\hat{\boldsymbol{z}}^{\mathrm{ref}}_{\mathrm{pre}}, \hat{\boldsymbol{z}}^{\mathrm{ref}}_{\mathrm{post}}$ for pre- and post-disturbance vectors, thus

$$\hat{\boldsymbol{z}}^{\mathrm{ref}} := [\hat{\boldsymbol{z}}^{\mathrm{ref}}_{\mathrm{pre}}, \ \hat{\boldsymbol{z}}^{\mathrm{ref}}_{\mathrm{post}}]^{\mathrm{T}} \in \mathbb{R}^Y.$$

Finally, the growth loss measures the distance between pseudo-labels and predictions, i.e.

$$\mathcal{L}_{\mathrm{growth}}(\boldsymbol{z}^{\mathrm{ref}}, \boldsymbol{z}^{\mathrm{pred}}) := \frac{1}{Y} ||\hat{\boldsymbol{z}}^{\mathrm{ref}} - \boldsymbol{z}^{\mathrm{pred}}||. \tag{2}$$

### 3.4 MODEL TRAINING

Training a spatio-temporal model at global scale requires careful design of the dataset construction and optimization strategy. The large size and geographic diversity of the input data demand a sampling strategy that balances coverage of relevant forested areas with computational feasibility, both for training and global-scale inference. Further, the sparse and (temporally and spatially) noisy nature of GEDI supervision necessitates specialized training objectives and stable optimization.

**Dataset.** Building on the multi-sensor inputs described in Section 3.1, we assembled a large-scale training dataset by sampling spatio-temporal patches centered on GEDI footprints. From each of the 13.000 Sentinel-2 tiles over land with GEDI coverage, we generated up to 230 patches depending on the availability of valid GEDI labels. In non-forested regions with limited relevance (e.g., Sahara) we restricted the number of patches to a maximum of three to reduce computational overhead. The resulting dataset contains approximately 3 million multi-sensor samples, totaling approx. 50 TB of input data. For model testing, we selected one hold-out sample per Sentinel-2 tile, ensuring broad spatio-temporal coverage across continents and biomes.

**Training Procedure.** We trained our model on 8 NVIDIA H200 GPUs for about one week with the hyperparameters detailed in Table 6 in the Appendix A.4. We first pretrained the model using the Huber loss on the reference head for 400k iterations with a batch size of 16. Subsequent finetuning for 47k iterations and a batch size of 8 was performed by training the prediction head with the growth loss detailed in Section 3.3, while freezing the rest of the model parameters.

## 4 RESULTS

We evaluate ECHOSAT through a comprehensive three-part assessment. Our evaluation uses GEDI labels filtered according to the quality criteria described in Appendix A.2.1, ensuring a reliable ground truth. We first assess prediction accuracy against GEDI labels, then analyze the temporal dynamics and growth patterns captured by our model, and finally compare our 2020 predictions against existing single-year baselines. When not specified otherwise, reported metrics exclude labels below $5\,\mathrm{m}$ following Hansen et al. (2013a), which defines trees as vegetation exceeding $5\,\mathrm{m}$ height.

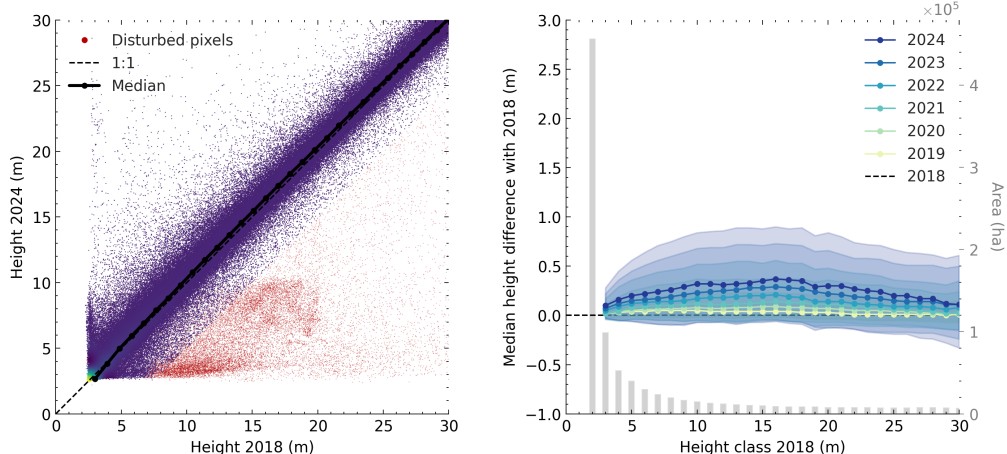

Figure 2: **Left.** Scatter plot showing the predicted height in 2018 against 2024. Disturbed pixels are identified by a decrease of more than $5\,\mathrm{m}$ between 2018 and 2024, marked red and excluded from the median aggregation. **Right.** Median height difference from 2018 to each year, binned in $1\,\mathrm{m}$ height classes. The right y-axis shows the height class distribution and area for these classes.

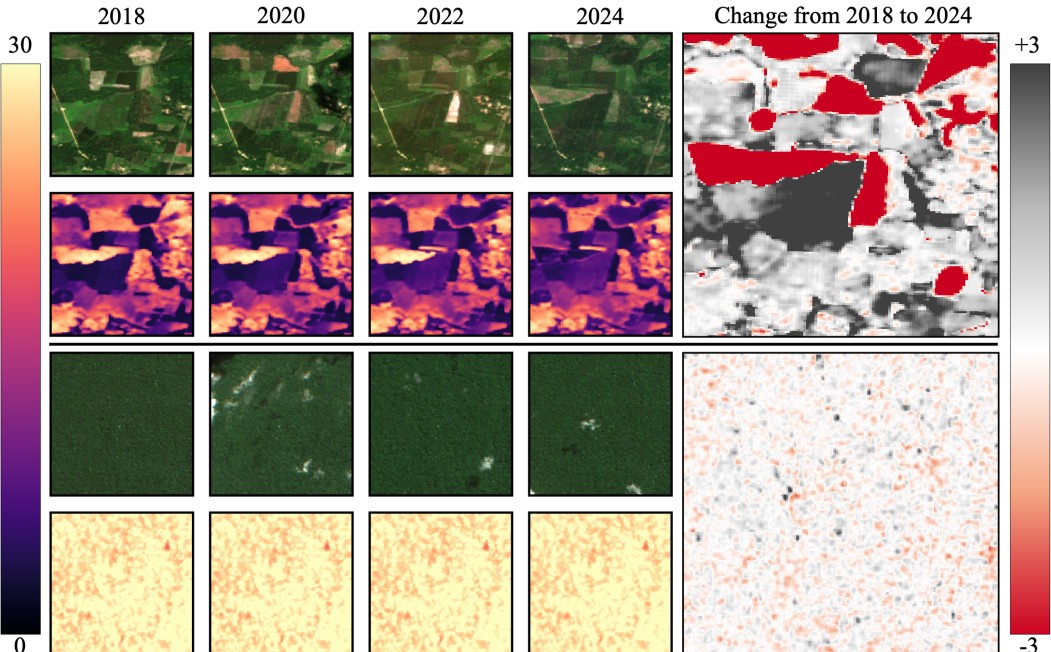

Figure 3: Examples of predicted tree height dynamics for two contrasting regions. **Top**: Le Landes (France) showing disturbance and regrowth patterns. **Bottom**: Amazonas (Brazil) with largely stable forest structure. Each block shows optical imagery (top row), predicted tree height (second row), and corresponding change maps from 2018 to 2024 (right column).

## 4.1 CANOPY HEIGHT ACCURACY

For 2019-2022, MAE values range from $5.36\,\mathrm{m}$ to $6.27\,\mathrm{m}$, indicating similar prediction accuracy across years. However, 2023-2024 show notable variations: MAE increases to $5.79\,\mathrm{m}$ in 2023, then decreases significantly to $4.89\,\mathrm{m}$ in 2024. This pattern correlates with GEDI's operational status, as the instrument was inoperational from March 17, 2023, through April 22, 2024, resulting in different label distributions and availability patterns. Details in Table 4 in Appendix A.3

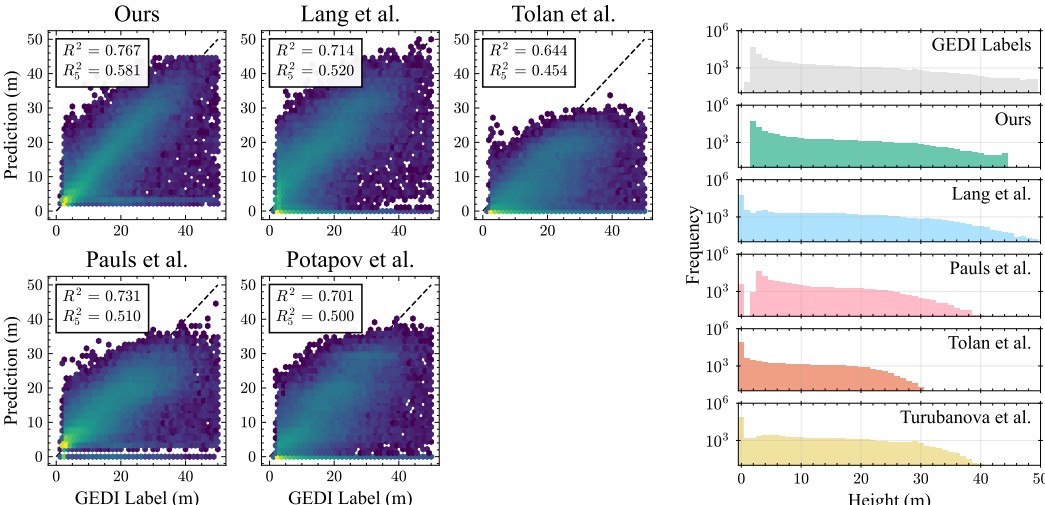

Figure 4: Left: Scattterplots showing the predicted height for 2020 vs GEDI labels with the correlation coefficient ($R^2$) and the correlation coefficient for labels exceeding $5\,\mathrm{m}$ ($R_5^2$) indicated for each plot. Right: Histogram of the (predicted) values.

The substantial gap between MAE ($4.89\,\mathrm{m}-6.27\,\mathrm{m}$) and RMSE ($8.59\,\mathrm{m}-11.21\,\mathrm{m}$) indicates the presence of large prediction errors, suggesting that while most predictions are reasonably accurate, occasional severe errors occur. This error distribution likely stems from remaining noise in GEDI labels after filtering, particularly cases where LiDAR waveforms fail to penetrate dense canopies, resulting in ground-level measurements ($0\,\mathrm{m}$) for trees that may actually exceed $30\,\mathrm{m}$ in height.

### 4.2 Canopy Height Growth/Decline

Due to the sparse temporal and spatial distribution of GEDI labels, a temporal validation with GEDI is not possible. Instead, we focus on analyzing the temporal dynamics captured by our model to assess whether the predictions exhibit realistic forest growth patterns.

The left part of Figure 2 presents a scatterplot of predicted heights in 2018 versus 2024, with median values plotted for each $1\,\mathrm{m}$ height bin. Pixels are marked disturbed when the height decreaes by more than $5\,\mathrm{m}$ over the time span. The right part of Figure 2 shows median height differences and lower and upper quartile for each $1\,\mathrm{m}$ height class from 2018 to each subsequent year, demonstrating year-to-year growth variations. The analysis reveals consistent growth across all height classes, with taller trees exhibiting slower growth rates, consistent with established forest growth patterns. Figure 3 shows the predictions and change for two areas: Highly active forests in Le Landes (France) and Amazonas rainforest (Brazil). In the Le Landes forest in France, which is well known for its intensive wood production and therefore fast-growing tree species, the predictions reveal many disturbances — most likely caused by logging activities — and phases of regrowth. In contrast, the predictions for the Amazonas region remain largely stable, showing very little variation across the different years. Satellite images from 2018 to 2024 together with time-series of pixel-wise predictions for five selected pixels around a disturbance in Le Landes are depicted in Figure 9 in Appendix A.3. Please note that our model is able to predict consistent canopy height over time, even for the first year 2018, where GEDI labels are not available.

### 4.3 Comparison against Existing Maps

While no global-scale temporal tree height maps exist, four single-year approaches provide suitable baselines for comparison: Tolan et al. (2024) (DINOv2-based, $1\,\mathrm{m}$ resolution), Potapov et al. (2021) (Random Forest, $30\,\mathrm{m}$ Landsat), Lang et al. (2023) (CNN, $10\,\mathrm{m}$ Sentinel-2), and Pauls et al. (2024) (UNet, $10\,\mathrm{m}$ Sentinel-2). We compare our 2020 predictions against these baselines using the same MAE, MSE, RMSE, MAPE and two $R^2$ metrics on our test samples. All baseline maps were downloaded from Google Earth Engine, rescaled to $10\,\mathrm{m}$ using bilinear interpolation, and warped

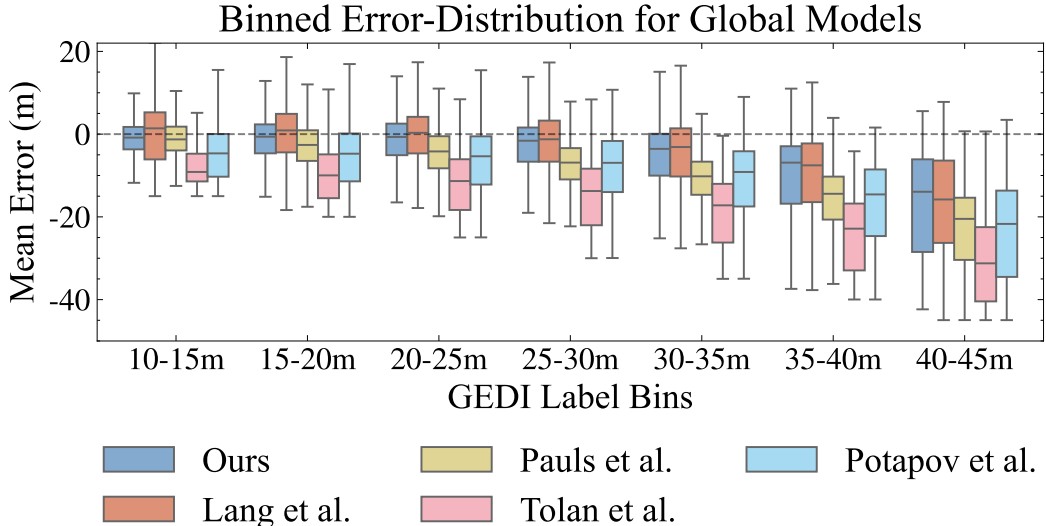

Figure 5: Error distribution analysis across height classes (5 m bins) for all baseline methods. Box-plots show mean absolute error for each height class, revealing that tall tree prediction remains challenging across all approaches, with errors increasing substantially for heights above 25 m.

to the corresponding Sentinel-2 tile CRS. Table 1 reports the quantitative comparison, showing that ECHOSAT consistently outperforms the other maps, while also reducing the variance in its predictions.

Figure 6 shows a visual comparison of all maps in 3 distinct regions. Although the map by Tolan et al. (2024) is resampled to 10 m, it visually still has a higher resolution and can be used very well for the detection of smaller tree patches. The map by Potapov et al. (2021) uses 30 m Landsat data as input and therefore the map fails to identify some trees, however the accuracy and tree height labels is better. Lang et al. (2023), Pauls et al. (2024) and our model use Sentinel-2 as input and can detect most smaller forest patches, but also have a higher accuracy on tree height labels. Pauls et al. (2024) and our model show finer structure in the prediction.

The scatterplots and histograms in Figure 4 and 5 reveal that Tolan et al. (2024), Pauls et al. (2024) and Potapov et al. (2021) saturate between 30 m and 35 m, while Lang et al. (2023) and our map can predict beyond that. As already indicated by the correlation coefficient, also the body of our scatterplot is narrower than the one by Lang et al. (2023). Although our predictions stop at roughly 45 m, comparing them to the GEDI distribution reveals a closer match than for Lang et al. (2023). Further figures are provided in Appendix A.3).

## 5 CONCLUSION

Our approach addresses the fundamental limitation of existing static forest height products through a novel growth loss framework that inherently enforces physically realistic forest dynamics without requiring post-processing. By leveraging multi-sensor satellite data and our Temporal-Swin-Unet, we demonstrate how temporal forest monitoring can be achieved at unprecedented scale and resolution. Our evaluation shows that different height classes of trees have varying growth rates, consistent with existing literature. On a single-year evolution comparing our map to other existing ones we show strong performance and improved accuracy in all evaluated metrics. This work provides essential capabilities for climate change mitigation, carbon accounting, and forest disturbance assessment, advancing our ability to monitor and understand global forest dynamics. The produced maps will be made available upon acceptance.

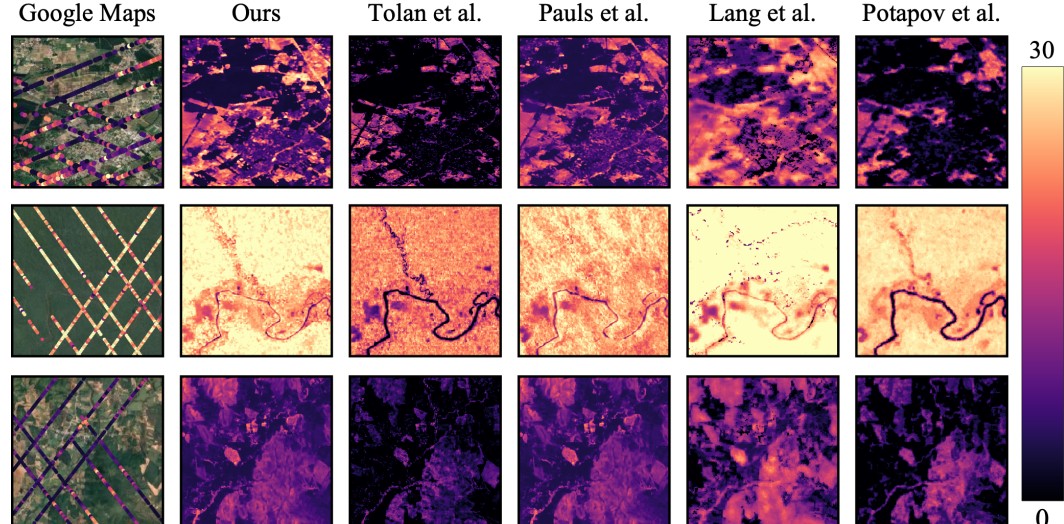

Figure 6: Qualitative comparison across three geographically diverse locations. The first column shows Google Maps imagery for spatial context, while subsequent columns display predicted tree heights ($0\,\mathrm{m}-30\,\mathrm{m}$ range) for each method. This visual assessment reveals differences in spatial detail, forest boundary detection, and height estimation accuracy across the various approaches.

## 5.1 LIMITATIONS

The proposed ECHOSAT maps have several limitations, which we sketch here:

**Ground truth constraints.**  GEDI labels exhibit systematic noise in cloudy regions (e.g., tropical rainforests) and for trees exceeding 50 m, where LiDAR signals fail to accurately reach the ground. GEDI cannot reliably measure vegetation below 5 m, limiting predictions for shrubs and small trees. The sparse spatial coverage ( 25 m footprints at irregular spacing) means most pixels lack direct supervision, requiring substantial spatial generalization. As GEDI began operations in late 2018, predictions for that year lack corresponding training labels.

**Detecting growth and disturbances.**  Smaller annual growth increments often fall within model and data uncertainty, making small height changes difficult to detect reliably. Our fixed disturbance criteria effectively capture clear-cuts and major fires but may miss gradual degradation, selective logging, or scattered tree mortality.

**Validation.**  GEDI's sparse temporal coverage does not allow direct validation of year-to-year height changes. While Section 4.2 shows our predictions follow realistic ecological growth patterns, validation against independent repeated observations (airborne LiDAR campaigns, permanent forest plots) is needed. The seven-year timespan (2018–2024) limits assessment of long-term dynamics and multi-decadal carbon accumulation patterns.

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

# A APPENDIX

## A.1 USE OF LARGE LANGUAGE MODELS

Large language models were used to aid in writing (polishing text), generating code for plots, and implementing standard components. No novel research ideas or results were produced by LLMs.

## A.2 METHODOLOGY

### A.2.1 DATA

Here we describe the used data sources and their processing in more detail. Table 2 provides an overview.

**Sentinel-2 Optical Data.** We use all 12 spectral bands from Sentinel-2 L2A products, which provide atmospheric correction and cloud probability estimates. For each year, we select one image per calendar month based on the highest percentage of valid pixels (excluding cloudy and black pixels as identified by the Sen2Core algorithm). The Sentinel-2 $10\,\mathrm{m}$ bands serve as the foundation for our dataset, hence bands with $20\,\mathrm{m}$ and $60\,\mathrm{m}$ native resolution are upsampled to $10\,\mathrm{m}$ using nearest neighbor interpolation. Values are normalized to the range [-1, +1] using band-specific scaling factors: bands 1-4 scaled from [0, 2000], bands 6-9 from [0, 6000], band 0 from [0, 1000], and bands 5, 10-11 from [0, 4000].

**Sentinel-1 Radar Data.** We utilize C-band synthetic aperture radar data with $10\,\mathrm{m}$ spatial resolution, processing quarterly median composites of VH polarization for both ascending and descending orbits. Digital numbers are converted to backscatter coefficients (dB) and scaled from [-50, +1] to [-1, +1]. The downloaded images from Google Earth Engine are already geospatially aligned to the $10\,\mathrm{m}$ Sentinel-2 bands, hence can just be stacked. To align with monthly Sentinel-2 data, each quarterly composite is duplicated across the corresponding three months.

**ALOS PALSAR-2 Radar Data.** We incorporate L-band synthetic aperture radar data with $30\,\mathrm{m}$ spatial resolution, using yearly median composites of HH and HV polarizations. Digital numbers are converted to backscatter coefficients (dB) and scaled from [-50, +1] to [-1, +1]. After reprojection and upsampling using bilinear interpolation with the pixels being aligned to the Sentinel-2 $10\,\mathrm{m}$ bands (using `gdalwarp`'s `-tap` option), the yearly composite is duplicated across all 12 months to maintain temporal consistency.

**Tandem-X Data.** We utilize two products from the TanDEM-X mission: (1) a $12\,\mathrm{m}$ resolution digital elevation model scaled from [0, 7000m] to [-1, +1], and (2) a forest/non-forest classification map with 3 classes normalized to [-1, +1]. Both products are duplicated across all 84 time steps (7 years × 12 months) as they represent static features.

**GEDI LiDAR Ground Truth.** We use GEDI L2A V2 products as ground truth labels, applying quality filters to ensure data reliability: relative height at 98th percentile (rh98) between $0\,\mathrm{m}-150\,\mathrm{m}$, only high-power beams, number of detected modes $>= 1$, quality flag = 1, degrade flag = 0, and sensitivity $>= 0.95$. These measurements provide sparse tree height estimates with approximately $25\,\mathrm{m}$ diameter footprints. Labels are rasterized to the Sentinel-2 $10\,\mathrm{m}$ bands using the provided geolocation of the highest return (given by `lon_highestreturn` and `lat_highestreturn`). In the very rare occurence of multiple GEDI labels having the same pixel and year associated, the maximum of them is taken. Although the GEDI diameter is often referred to as $25\,\mathrm{m}$, we just use the center pixel, as the returned energy decreases rapidly when distancing from the center (see Figure 7).

To ensure training data quality and focus on forested areas, we apply additional spatial filtering using Tandem-X data. We calculate terrain slope within a $70\,\mathrm{m}$ radius around each GEDI measurement and exclude locations with slopes exceeding $20°$ to avoid bare mountain areas. Additionally, we use the Tandem-X Global Urban Footprint to remove measurements where human footprint exceeds $10\%$, ensuring our model trains on natural forest environments rather than urban areas.

### A.2.2 TRAIN/TEST-SPLIT

Our testing dataset is created by randomly picking one area ($960\,\mathrm{m} \times 960\,\mathrm{m}$) inside each Sentinel-2 tile. To limit the effect of spatial autocorrelation, we enforce a minimum distance between training

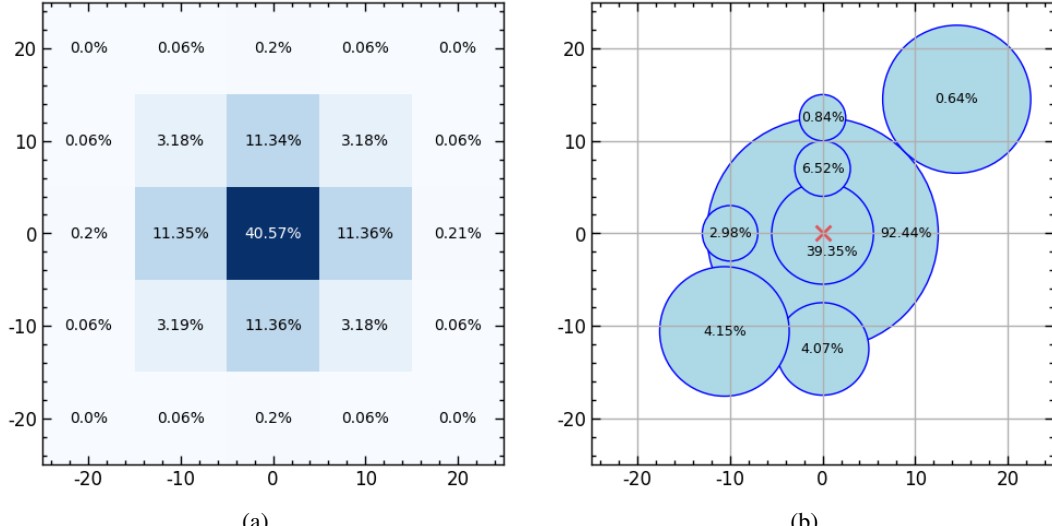

(a)                                    (b)

Figure 7: GEDI shot photon distribution. (a) The photon density of a single GEDI shot follows a normal distribution across both spatial dimensions. A Sentinel-2 pixel located directly at the center of the footprint receives 40.57% of the emitted laser energy. (b) Percentage of emitted photons received by sample trees at different positions. A tree with a $5\,\mathrm{m}$ radius centered directly under the GEDI footprint receives approximately 40% of the photons, while a tree of $7\,\mathrm{m}$ radius offset by $15\,\mathrm{m}$ from the center receives only 4.15%. Note that these values represent emitted photon distribution, not the reflected energy measured by the sensor. Actual measurements depend on material reflectance properties, ground return proportion, and the `rh98` metric extraction, requiring substantially more than 2% photon coverage for counting a tree as the largest tree in a GEDI measurement.

Table 2: Data sources used in ECHOSAT with their specifications and preprocessing parameters.

| Source | Origin | Res. | CRS | Resampling | Bands/Channels | Normalization |
|---|---|---|---|---|---|---|
| Sentinel-1 | Google Earth Engine | 10m | UTM | NN | VH (Asc/Des) | [-50, 1] in dB |
| Sentinel-2 | AWS | 10m | UTM | N/A | All w/o B10 | [0, 1000]: 0
[0, 2000]: 1–4
[0, 6000]: 6–9
[0, 4000]: 5/10/11 |
| ALOS Palsar-2 | JAXA FTP | 25m | EPSG:4326 | Bilinear | HH/HV | [-50, 1] in dB |
| TandemX EDEM | DLR Geoservice | 30m | EPSG:4326 | Bilinear | — | [0, 7000] |
| TandemX FNF | DLR Geoservice | 50m | EPSG:4326 | NN | — | [0–2] |
| GEDI L2A | NASA Earthdata | ~25m | EPSG:4326 | N/A | rh98 | — |

and testing patches of at least $360\,\mathrm{m}$. Figure 8 shows an analysis of spatial autocorrelation for our dataset, where the correlation is given by the Pearson product-moment correlation coefficient. The correlation starts at $\approx 0.74$ and reaches its sill at a lag between $300\,\mathrm{m}$ and $400\,\mathrm{m}$ at $0.55$.

### A.2.3   MODEL ARCHITECTURE

Here, we define the model architecture and design decisions in more detail.

**Patch Embed.** We implement the Patch Embed via a Conv3D layer, with kernel size and stride set to $(1, 1, 1)$, which is equivalent to applying a linear layer channel-wise for every pixel and every timestep. Embedding patches of size $4 \times 4$ pixels, as most other contemporary approaches do it, would lead to the model producing blurry forest borders and overlooking individual or extraordinarily tall trees.

**Temporal Downsample (TD) layer.** The TD layer takes as input a tensor of shape $T_{\mathrm{in}} \times H_{\mathrm{in}} \times W_{\mathrm{in}} \times E_{\mathrm{in}}$ and outputs a tensor of shape `reduce_time[`$l_{\mathrm{enc}}$`]` $\times H_{\mathrm{in}}/2 \times W_{\mathrm{in}}/2 \times 2E_{\mathrm{in}}$. In our

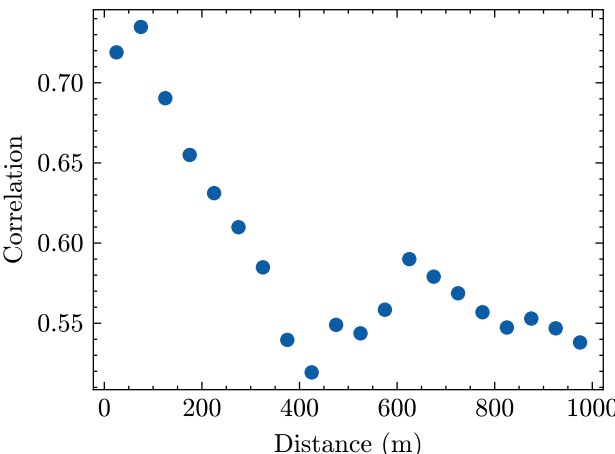

Figure 8: Analysis of spatial autocorrelation between GEDI labels in our dataset. Correlation is given by the Pearson product-moment correlation coefficient in bins of $50\,\mathrm{m}$. The correlation starts to level off at around $300\,\mathrm{m}$ ot $400\,\mathrm{m}$.

model, we set `reduce_time = [28, 14, 7]`, so the time dimension is reduced from $84$ to $28$ by a factor of three in Encoder layer 1, and then twice by a factor of two. The temporal reduction is implemented via a linear layer (without a bias) applied individually for every year and pixel by concatenating all embeddings of a pixel of a given year, then linearly projecting it down to the target temporal resolution. Afterwards, the spatial resolution is halved by concatenating the embeddings of four spatially adjacent pixels, applying Layer Normalization (Ba et al., 2016) and then applying another linear projection. The layer's output is the input for the following Encoder layer, and for the TSC layer of the corresponding Decoder layer via a skip connection.

**Temporal Skip Connection (TSC) layer.** The TSC layer is the Decoder-counterpart of the TD layer. Note how the time dimension changes throughout the model: it is iteratively reduced from $84$ to $7$ in the Encoder, but stays $7$ throughout the whole Decoder, as we need a single prediction map per year. This prohibits the simple addition of Encoder and Decoder inputs in the TSC layer. After trying out multiple designs, we settled on a Transformer layer, which we will now explain in detail for the skip connection between Encoder layer 1 and Decoder layer 4. The input coming from Encoder layer 1 has shape $84 \times H \times W \times E$ and the output of Decoder layer 4 has shape $7 \times H \times W \times E$. Now, we reshape and concatenate these inputs into a tensor of shape $7HW \times \frac{84+7}{7} \times E$. For every pixel and year (= voxel), we have 13 features, one from the decoder, the rest from the encoder. The decoder feature of a pixel can thus attend to its encoder features of the same year, before being passed on. After the attention we only keep the decoder token and ignore the others.

**3D Window Multi-Head Self Attention (3D W-MSA).** In the 3D window multi-head self attention, each token can attend to the other tokens within the same window, which spans two tokens along the temporal dimension and six along both spatial dimensions. In typical Video Swin Transformer fashion, every other attention block is shifted in time and space dimension. Thus, every token can attend to the 71 other tokens in the same window, 36 of which are from the previous or following timestep. Using an efficient attention mechanism is necessary due to the otherwise quadratic complexity in the number of pixels $HW$ in an image. Windowed attention is a strong contender, inducing a locality bias while sacrificing the global receptive field in return. In subsequent Encoder layers, the receptive field of the windowed attention is progressively doubled, due to the downsampling operations.

**Conv Heads.** The reference head consists of a single 3D convolution that projects linearly from the embedding dimension to a scalar per voxel. The prediction head starts with two 3D convolutions with kernel size 3 in temporal and spatial dimension and keeping the embedding dimension unchanged, followed by Group Normalization (Wu & He, 2018) and ReLU activation. Therefore, neighboring pixels and consecutive years are able to interact with each other. The final step is a single 3D convolution as in the reference head. The prediction head is designed to allow local spatial and temporal interaction, in order to facilitate the precise detection of forest borders or disturbance

Table 3: Ablation: Comparison of our approach with (GrowthLoss and a simple linear regression) and without fine-tuning for labels from 2020 and above $5\,\mathrm{m}$ regarding MAE (m), MSE ($\mathrm{m}^2$), RMSE (m), MAPE (%), $R^2$ and $R^2_{\mathrm{all}}$ (on all labels, including labels above $5\,\mathrm{m}$). IQR is given in square brackets. The pseudo-label fine-tuning with our proposed Growth-Loss (see Section 3.3) does not affect the accuracy regarding the reference labels

| Method | MAE ↓ | MSE ↓ | RMSE ↓ | MAPE ↓ | $R^2$ ↑ | $R^2_{\mathrm{all}}$ ↑ |
|---|---|---|---|---|---|---|
| Ours (w/ GrowthLoss) | 5.85 [4.90] | **118.07** [36.16] | **10.87** [4.90] | 30.20 [33.32] | **0.59** | **0.77** |
| Ours (w/ Linear) | 5.87 [4.89] | 118.51 [36.14] | 10.89 [4.89] | 30.34 [33.36] | **0.59** | **0.77** |
| Ours (w/o fine-tuning) | **5.84** [4.89] | 119.36 [35.55] | 10.93 [4.89] | **29.44** [32.13] | **0.59** | **0.77** |

locations. Without this interaction, our models tend to have problems with border detection, presumably due to geolocation uncertainty of GEDI measurements.

### A.3 RESULTS

#### A.3.1 ABLATION: FINE-TUNING

We present an ablation study of our fine-tuning methodology, comparing the model without fine-tuning, fine-tuning with our GrowthLoss, and fine-tuning with a simple linear regression. Table 3 shows that all three approaches yield similar results in terms of quantitative metrics, which is expected since fine-tuning only uses the pseudo-labels produced by the pretrained backbone. The impact of fine-tuning becomes evident when examining the temporal evolution of pixel predictions (Figure 9). Without fine-tuning, the model can capture both tree growth and cutting events, but exhibits unwanted height fluctuations. Linear regression fine-tuning smooths tree growth predictions but fails to properly capture disturbances. Fine-tuning with our GrowthLoss, in contrast, reduces uncertainty in growth pixels, detects disturbances more accurately, and minimizes height fluctuations in disturbance border areas.

#### A.3.2 YEAR-WISE EVALUATION

Table 4 presents MAE, MSE, RMSE, MAPE, and two $R^2$ metrics for our ECHOSAT maps for each year individually. Errors show a gradual decrease over time, which may suggest that recent years are easier to predict, but could also reflect improvements in GEDI label processing by NASA. A similar pattern is observed when examining errors across different height bins for each year (Figure 10).

Table 4: Year-wise comparison on error metrics regarding MAE (m), MSE ($\mathrm{m}^2$), RMSE (m), MAPE (%), $R^2$ and $R^2_{\mathrm{all}}$ (on all labels, including labels below $5\,\mathrm{m}$). IQR is given in square brackets.

| Year | MAE ↓ | MSE ↓ | RMSE ↓ | MAPE ↓ | $R^2$ ↑ | $R^2_{\mathrm{all}}$ ↑ |
|---|---|---|---|---|---|---|
| 2019 | 6.09 [5.25] | 123.44 [40.48] | 6.09 [5.25] | 0.30 [0.34] | 0.59 | 0.77 |
| 2020 | 5.85 [4.90] | 118.07 [36.16] | 5.85 [4.90] | 0.30 [0.33] | 0.59 | 0.77 |
| 2021 | 5.41 [4.71] | 96.03 [33.49] | 5.41 [4.71] | 0.30 [0.33] | 0.63 | 0.79 |
| 2022 | 5.24 [4.75] | 85.37 [34.34] | 5.24 [4.75] | 0.29 [0.31] | 0.66 | 0.82 |
| 2023 | 5.56 [4.94] | 102.86 [35.77] | 5.56 [4.94] | 0.29 [0.31] | 0.62 | 0.79 |
| 2024 | 4.77 [4.26] | 72.01 [27.78] | 4.77 [4.26] | 0.28 [0.28] | 0.68 | 0.83 |

#### A.3.3 SPATIAL DISTRIBUTION OF ERRORS

Figure 11 (a) shows a global map where the average error in each region is indicated by color (darker colors correspond to higher errors). Because height and error are closely correlated (as seen in Figure 5 and Table 5 for labels $< 5\,\mathrm{m}$), this map primarily reflects the average tree height per region. Figures 11 (b–g) show the same map broken down into $5\,\mathrm{m}$ height bins. While errors generally increase with height, the relatively small errors for large trees in the Amazonas rainforest and Congo Basin may indicate lower structural heterogeneity in these forests, making them easier to predict.

Table 5: Comparison of global-scale methods for 2020 only for labels below $5\,\mathrm{m}$ regarding MAE (m), MSE ($\mathrm{m}^2$), RMSE (m), MAPE (%), $R^2$ and $R^2_{\mathrm{all}}$ (on all labels, including labels above $5\,\mathrm{m}$). Please note that while GEDI measures ground elevation between $2\,\mathrm{m}$ and $3\,\mathrm{m}$, Tolan et al. (2024); Lang et al. (2023); Potapov et al. (2021) either use other labels or manually set their ground predictions to $0\,\mathrm{m}$, thereby making a comparison here ineffective.

| Method | MAE ↓ | MSE ↓ | RMSE ↓ | MAPE ↓ | $R^2$ ↑ | $R^2_{\mathrm{all}}$ ↑ |
|---|---|---|---|---|---|---|
| Potapov et al. (2021) | 2.84 [0.56] | 9.08 [3.08] | 3.01 [0.56] | 98.20 [0.00] | 0.26 | 0.70 |
| Lang et al. (2023) | 3.04 [0.71] | 13.30 [3.95] | 3.65 [0.71] | 104.08 [0.00] | 0.28 | 0.71 |
| Pauls et al. (2024) | 1.08 [0.96] | 2.99 [1.73] | 1.73 [0.96] | 37.04 [31.31] | **0.49** | 0.73 |
| Tolan et al. (2024) | 2.87 [0.60] | 8.93 [3.32] | 2.99 [0.60] | 99.04 [0.00] | 0.16 | 0.64 |
| Ours | **0.51** [0.42] | **1.82** [0.24] | **1.35** [0.42] | **15.99** [13.12] | 0.39 | **0.77** |

### A.3.4 LIDAR COMPARISON

Validating large-scale tree-height products remains challenging. GEDI provides global coverage, though with some noise and uncertainty, while airborne LiDAR (ALS) offers higher accuracy but is spatially sparse, biased toward well-surveyed regions, and rarely has temporal revisits suitable for year-to-year evaluation. For these reasons, GEDI remains the most practical benchmark for a global-scale study.

To provide additional context, we include an evaluation using ALS-derived labels in the Landes forest (France) from the LiDAR HD campaign (Figure 12). These results confirm local accuracy but are specific to this region and do not generalize globally. While ALS comparisons are valuable, systematic multi-region ALS validation is beyond the scope of this work.

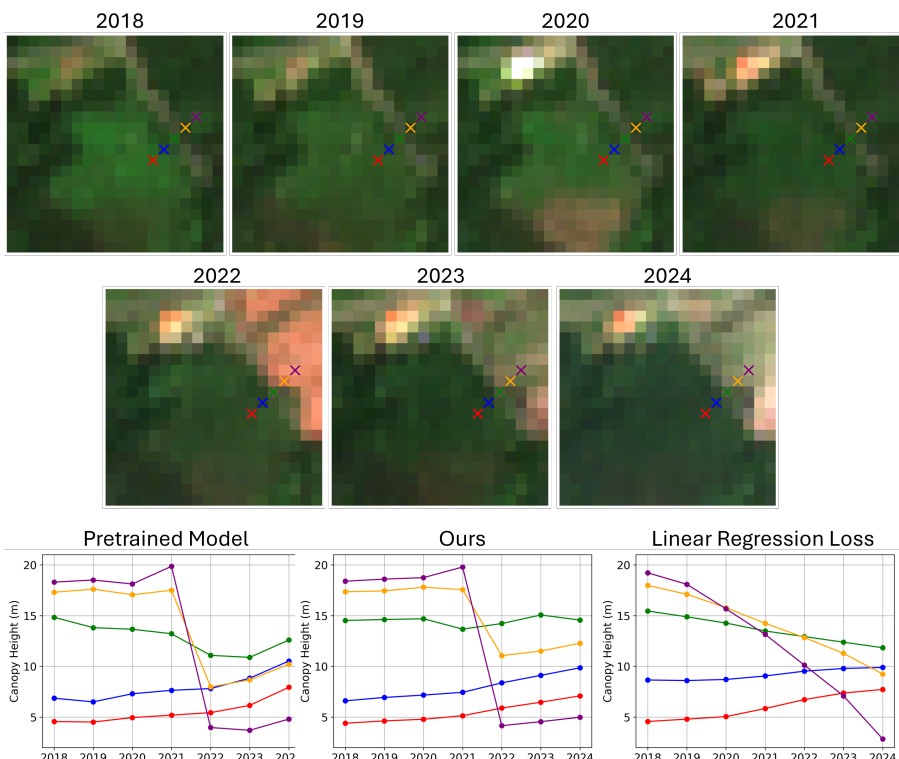

Figure 9: Sentinel-2 time series and temporal tree height prediction for five neighbouring pixels around a disturbance for the pretrained model (Left), our model (Middle) and model fine-tuned with a single linear regression model (Right). The simple linear regression model is fine-tuned on pseudo-labels produced by a linear interpolation of the pretrained model's predictions.

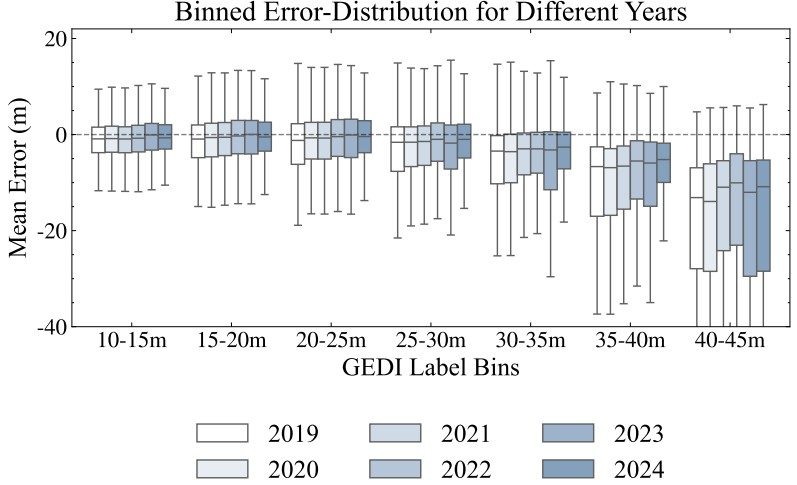

Figure 10: Error distribution analysis across height classes (5 m bins) for ECHOSAT predictions and different years. Boxplots show mean absolute error for each height class. The errors for smaller trees are very similar, with higher trees showing earlier years as being slightly more difficult than recent years.

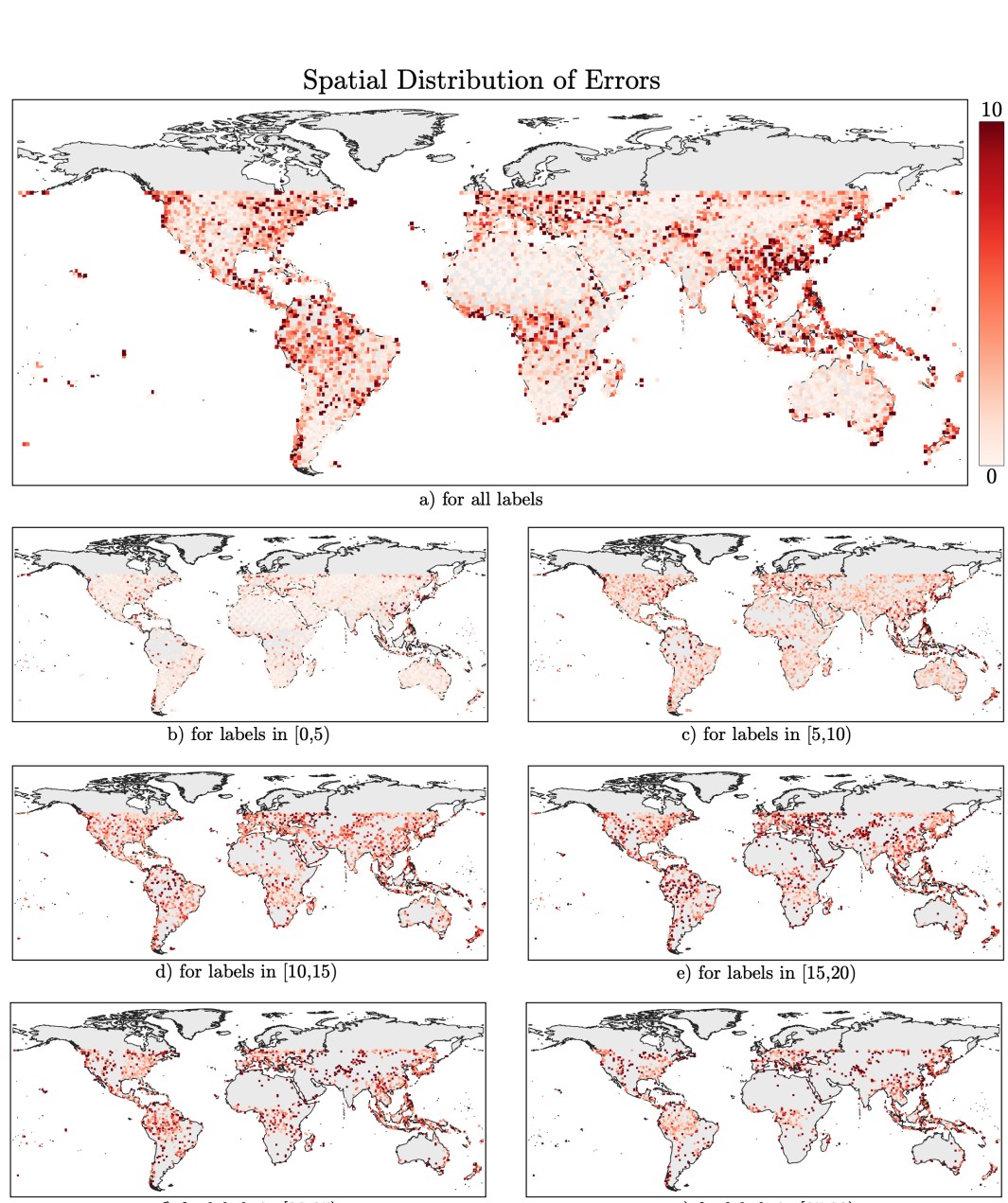

Figure 11: Spatial distribution of errors for all labels (a) and different height bins (b-g). a) strongly correlates with known areas of taller trees (i.e. higher errors, see Figure 5), small trees (b-c) generally have low errors globally, whereas errors systematically growth with tree height (d-g). Interestingly, (1) the east coast of North America has lower errors compared to other areas and (2) the Amazonas rainforest and Congo Basin, known for tall trees, have smaller errors (g) compared to other regions.

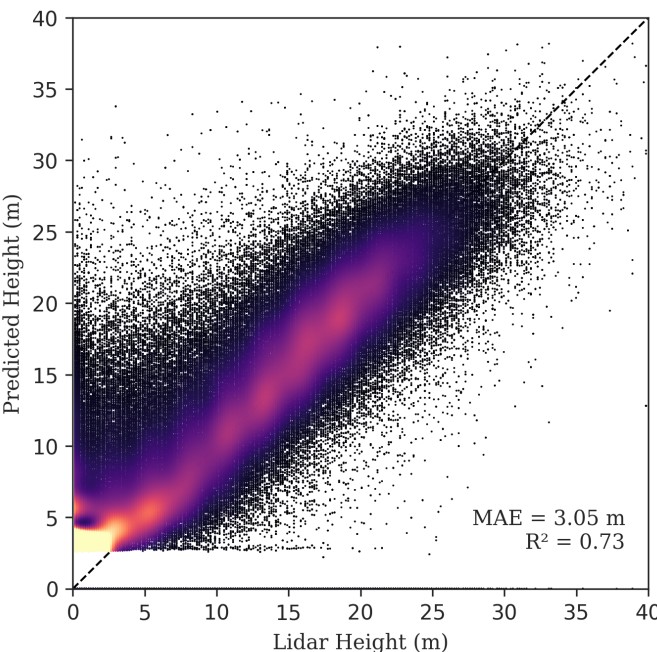

Figure 12: Density scatterplot between ALS tree height labels and ECHOSAT predictions in the Le Landes forest plantation area in France. LiDAR campaigns are from the LiDAR HD of the French Geoservices and have been reprojected and downsampled to 10m and EPSG:32630 using max pooling. The prediction is taken from the corresponding year of the ALS campaign. As our model is only trained with GEDI labels and GEDI labels cannot differentiate between small trees, bushes and ground elevation, it sets the minimum height to $\approx 3\,\text{m}$. Bright colors indicate higher density.

## A.4 MODEL HYPERPARAMETERS

Table 6: Hyperparameters used for model training. Some parameters differ between pretraining and finetuning, while most remain unchanged.

| Parameter | Symbol | Value | |
|---|---|---|---|
| | | *Pretraining* | *Finetuning* |
| Number of years | $Y$ | 7 | |
| Number of timesteps | $T$ | $84 \, (= Y \cdot Months)$ | |
| Number of input channels | $C$ | 18 | |
| Embedding dimension | $E$ | 72 | |
| Height / Width (in pixels) | $H / W$ | 96 | |
| Encoder depths | $\text{depth}[l_{\text{enc}}]$ | $[6, 4, 4, 6]$ | |
| Decoder depths | $\text{depth}[l_{\text{dec}}]$ | $[4, 6, 8, 16]$ | |
| Attention heads | | $[4, 8, 12, 24]$ | |
| Temporal window size | | 2 | |
| Spatial window size | | 6 | |
| Embedding patch size | $P_H \times P_W \times P_T$ | $1 \times 1 \times 1$ | |
| Optimizer | | `AdamW` | |
| Maximum learning rate | | $1 \times 10^{-4}$ | $3 \times 10^{-3}$ |
| Learning rate linear warmup | | 30% | |
| Learning rate schedule | | Cosine Annealing | |
| Gradient clipping | | 1 | |
| Number of iterations | | 400k | 47k |
| Loss | | $\mathcal{L}_{\text{huber}}(\cdot)$ | $\mathcal{L}_{\text{growth}}(\cdot)$ |
| Batch size | | 16 | 8 |

