# OpenReview forum: "ECHOSAT: Estimating Canopy Height Over Space And Time"
_ICLR.cc/2026/Conference — Submitted to ICLR 2026_

### Official Review · Reviewer_c1mT · 2025-10-24

**Soundness:** 1
**Presentation:** 2
**Contribution:** 1
**Rating:** 2
**Confidence:** 5

**Summary:**

The authors present ECHOSAT, the first spatio-temporal global-scale canopy height map at 10m resolution spanning seven years. They exploit multi-source satellite imagery to train a vision transformer that performs pixel-wise temporal regression with an adapted loss function designed for sparse temporal supervision. Analyses demonstrate that the model outperforms competing methods in single-date evaluations and that it learns realistic forest height dynamics over time.

**Strengths:**

1. The downstream application of canopy height estimation for large-scale biomass estimation is relevant and important for climate change understanding and mitigation.

2. Introducing optical and SAR time series for canopy height estimation at global scale and 10m resolution to include forest dynamics is a novelty and relevant for the task.

3. The design of the proposed Growth loss is a novelty and inspired by domain knowledge from the field of application. It accounts for potential forest disturbances, addresses missing LiDAR ground truth by filling gaps with regressions, employs constrained piecewise linear regressions to generate pseudo-labels, and combines real and pseudo-labels.

4. An analysis of canopy growth and decline is provided (Section 4.2, Figure 2) thanks to the integration of time series. This contribution is well appreciated and could open discussions for future work.

**Weaknesses:**

1. There is a clear lack of related work on: 1/ spatio-temporal methods for remote sensing that have shown to learn spatio-temporal dynamics on various tasks [1, 2, 3], and 2/ remote sensing foundation models pretrained on large-scale optical and SAR datasets [4, 5, 6], sometimes used for downstream forest monitoring applications [6, 7]. One may note that these methods could be leveraged either as a starting point for architecture design or for canopy height estimation fine-tuning.

2. Similarities and differences with other time series-based methods for forest monitoring remote sensing applications [8, 9, 10] have not been discussed.

3. There is a clear lack of ablation studies to understand whether the performance gain comes from the architecture selection or the loss function design. 1/ One would appreciate a comparison of the Growth loss in its final form with a simpler formulation using a single linear regression, and with losses from competing works such as standard MSE. 2/ There is no comparison with simple competing methods (e.g., U-Net) combined with the Growth loss. 3/ It would have been appreciated to compare other architectures to the Temporal-Swin-Unet to better understand its relative performance gain, such as a 3D U-Net [11] or remote sensing-based architectures as mentioned above.

4. The authors did not provide standard deviations of their quantitative results in Table 1, questioning the actual gain compared to competing methods. One may question the actual gain of the proposed method compared to Pauls et al. [12]: the proposed method achieves better performance on average, while absolute values of both methods are comparable, and Table 6 shows average errors per tree height that are similar within similar box plot interquartile ranges.

5. One would question the hypothesis of excluding labels below 5m according to Hansen et al. methodology [13] since: 1/ this guideline has not been followed by other competing methods, 2/ specific metrics for tree heights < 5m and > 5m could be easily defined to better distinguish use cases, and 3/ estimating tree heights < 5m is an actual use case for monitoring recent forest restoration projects through time. While all competing methods provide predictions < 5m (Table 4, right), it is not clear why the authors exclude this particular use case, whereas it represents a significant use case where the margin for improvement seems reasonable and would be useful.

6. There is a lack of explanation about the train, validation, and test split definitions that must be clarified to avoid significant issues. As an example, the methodology followed by Pauls et al. [12] is questionable since the splits have been defined by random patches that could introduce data leakage between the train and test sets through spatial autocorrelation.

7. The authors neither mention limitations of their work nor provide pathways to future work.

**References**:

[1] V. Sainte Fare Garnot & L. Landrieu, Panoptic Segmentation of Satellite Image Time Series with Convolutional Temporal Attention Networks. In ICCV 2021.

[2] M. Tarasiou et al., ViTs for SITS: Vision Transformers for Satellite Image Time Series. In CVPR 2023.

[3] G. Tseng et al., Lightweight, Pre-trained Transformers for Remote Sensing Timeseries. In ArXiv 2024.

[4] A. Fuller et al., CROMA: Remote Sensing Representations with Contrastive Radar-Optical Masked Autoencoders. In NeurIPS 2023

[5] G. Tseng et al., Galileo: Learning Global & Local Features of Many Remote Sensing Modalities. In ICML 2025.

[6] G. Astruc et al., AnySat: One Earth Observation Model for Many Resolutions, Scales, and Modalities. In CVPR 2025.

[7] N. Bountos et al., FoMo: Multi-Modal, Multi-Scale and Multi-Task Remote Sensing Foundation Models for Forest Monitoring. In AAAI 2025.

[8] T. Nguyen et al., Multi-temporal forest monitoring in the Swiss Alps with knowledge-guided deep learning. In Remote sensing of environment 2024.

[9] K. Wu et al., A semantic-enhanced multi-modal remote sensing foundation model for Earth observation. In Nature machine intelligence 2025.

[10] Z. Yu et al., QRS-Trs: Style Transfer-Based Image-to-Image Translation for Carbon Stock Estimation in Quantitative Remote Sensing. In EEEI Access 2025.

[11] O. Cicek et al., 3D U-Net: Learning Dense Volumetric Segmentation from Sparse Annotation. In MICCAI 2016.

[12] Pauls et al., Estimating Canopy Height at Scale. In ICML 2024.

[13] Hansen et al., High-resolution global maps of 21st-century forest cover change. In Science 2013.

**Questions:**

**Questions:**

1. What is the quantitative gain of the piecewise linear regressions versus using a single linear regression in the Growth loss function?

2. Is the model trained from scratch or from pretrained weights? If it is trained from scratch or pretrained on natural image-based datasets, why did the authors not consider exploiting various remote sensing backbones pretrained on large-scale optical and SAR datasets? Note that some of them are actually designed to exploit time series [1, 2, 3, 4].

3. Height estimation is a proxy for biomass estimation, which is the most important downstream application. In this work and related work, errors are mostly quantified via height estimation. However, what would be the equivalent of the height error in biomass estimation? Since the global allometric equation is not linear, this link is not straightforward and has been barely studied in previous canopy height map estimation works. Where are the biomass errors most important (small, medium, large trees) in average and absolute values?

4. Considering the increasing number of UAV LiDAR datasets [5, 6, 7], why did the authors not attempt to better evaluate methods on more precise datasets than GEDI-based annotations?


**Comments:**

• Please add numbers to relevant equations on page 5.

• One would appreciate integrating Figure 6 into the main paper, as the presented results are valuable and insightful.

• L. 353: "For 2019-2022, MAE values range from 5.36 m to 6.27 m, indicating consistent prediction accuracy." One may consider softening this claim since a 5-6m error on trees between 10-30m is quite significant for estimating their biomass.

• Section 4.1: One would appreciate an additional analysis of errors through time per tree height range, similar to Table 6.

• Figure 5: Please integrate a few GEDI LiDAR point clouds within the same color scale on the Google Map images to better understand the order of magnitude of the ground truth. With the current form of the figure, we can visually observe the difference in resolutions but cannot assess the quality of height predictions.


**References:**

[1] M. Tarasiou et al., ViTs for SITS: Vision Transformers for Satellite Image Time Series. In CVPR 2023.

[2] G. Tseng et al., Lightweight, Pre-trained Transformers for Remote Sensing Timeseries. In ArXiv 2024.

[3] G. Tseng et al., Galileo: Learning Global & Local Features of Many Remote Sensing Modalities. In ICML 2025.

[4] K. Wu et al., A semantic-enhanced multi-modal remote sensing foundation model for Earth observation. In Nature machine intelligence 2025.

[5] S. Puliti et al., FOR-instance: a UAV laser scanning benchmark dataset for semantic and instance segmentation of individual trees. In ArXiv 2023.

[6] B. Xiang et al., ForestFormer3D: A Unified Framework for End-to-End Segmentation of Forest  LiDAR 3D Point Clouds. In ICCV 2025.

[7] M. Wielgosz et al., SegmentAnyTree: A sensor and platform  agnostic deep learning model for tree  segmentation using laser scanning data. In Remote Sensing of Environment 2024.

---

> ### Author Response · Authors · 2025-11-21
> **Rebuttal (Part 1)**
>
> We thank the reviewer for the extensive and valuable feedback. Please see the official comment for a list of all changes made to the paper. Below, we address your feedback in detail:
>
> ### Model Training/Architecture CHoice
>
> We appreciate the reviewer's careful attention to recent advances in remote-sensing foundation models, and we fully understand the expectation that such models could provide a strong starting point. We revisited this point in depth while preparing the rebuttal.
>
> **Why not use existing foundation models?**
> We systematically evaluated all publicly available encoders and foundation models mentioned by the reviewer and several additional ones. While impressive and valuable to the community, each presented constraints that made them unsuitable for our specific task and operating scale:
>
> 1. Modality or granularity mismatch: Some models are pixel-based rather than image-based [3], or operate with patch sizes >1 [2], which do not capture spatial details essential for 10 m canopy-height regression.
> 2. Training-data scale limitations: Many models are trained on significantly smaller corpora (e.g., 1.5 TB [4], 150 GB [6], 150k training points [5]). In contrast, our task makes use of a 50 TB training corpus with 3M samples, and inference must efficiently scale to $\approx$ 1.5 PB of global data.
> 3. Architectural overhead not needed for our use case: Some models (e.g., [5]) introduce sophisticated multi-scale and multi-modal processing steps, which are powerful in general EO settings but also add unnecessary complexity and computational costs for our single-task and single-resolution regression setup. Given the global-scale deployment, even small inefficiencies quickly become prohibitive.
>
> **Why train from scratch?**
> Because our dataset is large and highly task-specific, training from scratch provides (i) alignment between model capacity and task requirements, (ii) predictable computational scaling, and (iii) more stable optimization than adapting a pretrained backbone with mismatched inductive biases or modalities. Also note that training is generally not the main bottleneck for the task at hand, in contrast to the inference phase, which is computationally more demanding.
>
> **Why Temporal-Swin-UNet?**
> None of the cited pretrained models, and no other SOTA EO backbones we reviewed (e.g., DOFA [101], DeCUR [102], SatMAE [103], SatMAE++ [104]), makes use of hierarchical feature learning as it is done by the Swin Transformer [105], which has consistently shown strong benefits for dense prediction tasks such as image-to-image regression. For our application, the hierarchical design provided the best balance of accuracy, efficiency, and memory footprint.
>
> **Framing our contribution.**
> Our focus is on tree-height estimation, and we ensured the related work section thoroughly reflects that domain. While general EO foundation models are an exciting direction, at the time of development, they did not yet provide a practical or computationally viable solution for accurate spatio-temporal canopy-height prediction at global scale.
>
> [101] Xiong et al. Neural plasticity-inspired foundation model for observing the earth crossing modalities. arXiv e-prints, pp. arXiv–2403, 2024.
>
> [102] Wang et al. Decoupling common and unique representations for multimodal self-supervised learning. In European Conference on Computer Vision, pp. 286–303. Springer, 2024.
>
> [103] Cong et al. Satmae: Pre-training transformers for temporal and multi-spectral satellite imagery. Advances in Neural Information Processing Systems, 35:197–211, 2022.
>
> [104] Noman et al. Rethinking transformers pre-training for multi-spectral satellite imagery. In Proceedings of the IEEE/CVF Conference on Computer Vision and Pattern Recognition, pp. 27811–27819, 2024.
>
> [105] Liu, Ze, et al. "Swin transformer: Hierarchical vision transformer using shifted windows." Proceedings of the IEEE/CVF international conference on computer vision. 2021.

---

> ### Author Response · Authors · 2025-11-21
> **Rebuttal (Part 2)**
>
> ### Architecture & Loss Ablation/Interplay
> This is an important point, and expanding the analysis helps to clarify the distinct roles of the backbone and the temporal loss. A key challenge is that GEDI provides almost no repeated measurements for the same pixel across years, which limits the feasibility of quantitative temporal validation. As a result, single-date metrics primarily reflect architectural performance, while temporal behavior must be assessed indirectly. We expanded the analysis along three axes:
>
> 1. **Model with vs. without fine-tuning (Table 3):** GrowthLoss has only a negligible effect on single-date accuracy; it mainly shapes temporal behavior.
> 2. **Simple linear regression baseline:** Replacing the GrowthLoss with a single linear fit alters the metrics only slightly, which is expected given that strong disturbances are relatively uncommon.
> 3. **Updated Figure 8:** Temporal consistency clearly degrades without fine-tuning or when using a simple linear regression. GrowthLoss produces the most coherent year-to-year evolution.
>
> These additions make explicit that the backbone determines general performance, while GrowthLoss is responsible for enforcing temporal consistency rather than improving point-wise accuracy.
>
>
> ### Metrics & Improvements
> This point is important to clarify. The model does not apply any 5 m threshold to GEDI labels during training or inference. All labels are used. The threshold appears only in the evaluation protocol, where we report a subset of metrics on labels > 5 m. The reason is that GEDI provides a large number of ground and near-ground returns, which are substantially easier to predict than true canopy heights. Including them in aggregated metrics would mask meaningful differences between methods and inflate apparent performance. Restricting certain metrics to > 5 m allows a fairer comparison focused on trees, which are the primary target of this work.
>
> Because GEDI's sampling characteristics make it difficult to reliably capture growth trajectories of trees below 5 m, a temporal analysis for this height range is inherently limited (as discussed in the added ``Limitations'' section). Still, to address the concern directly, we expanded the results:
>
> 1. Table 1 now includes interquartile ranges (IQR) to make performance differences easier to interpret.
> 2. A new Table 5 reports metrics for labels < 5 m, enabling a complete view of performance across both small and tall vegetation.
> 3. We have added Figure 6 (now 5) to the main paper and added a similar figure for single years to the appendix.
>
> ### Train/Test Split
> Spatial autocorrelation can be a problem if training and test samples are too close to each other. To avoid this, we keep a minimum distance of 360 m between all training and test points. This gap makes sure that no test point is influenced by nearby training data and keeps the evaluation independent.
>
> ### Limitations & Future Work
>
> We have added Section 5.1, which now details the limitations of our work.
>
>
> ### Height Errors and Above-Ground Biomass (AGB) propagation
>
> Above-Ground Biomass (AGB) is usually modeled as a power-law of tree height, for example:
>
> $$\text{AGB} = a \cdot H^b$$
>
> where (a) and (b) are coefficients fitted to the forest type and region. This means that even an error in height of 1 m can translate into a non-linear error in AGB, and the impact depends on the tree height.
>
> Different regions and forest types have different coefficients, so local calibrations usually outperform global ones (see, e.g., Saatchi et al. [106], Figure 1). This highlights that height errors do not map linearly to biomass errors and that the effect is context-dependent.
>
> [106] Saatchi, Sassan S., et al. "Benchmark map of forest carbon stocks in tropical regions across three continents." Proceedings of the national academy of sciences 108.24 (2011): 9899-9904.
>
> ### LiDAR comparison
> Validating large-scale tree-height products remains challenging. GEDI provides global coverage, though with some noise and uncertainty, while airborne LiDAR (ALS) offers higher accuracy, but is spatially sparse, biased toward well-surveyed regions, and rarely has temporal revisits suitable for a year-to-year evaluation. For these reasons, GEDI remains the most practical benchmark for a global-scale study.
>
> To provide additional context, we include an evaluation using ALS-derived labels in the Landes forest (France) from the LiDAR HD campaign (see Figure 10 in the appendix). These results confirm local accuracy, but are specific to this region only and do not generalize globally. While ALS comparisons are valuable, a systematic multi-region ALS validation is beyond the scope of this work.
>
> Thank you again for your feedback. We hope to have addressed your concerns and updated our manuscript accordingly. If there is any point you would like us to clarify or discuss in more depth, we would be happy to do so.

---

> > ### Comment · Reviewer_c1mT · 2025-11-24
> >
> > I would like to respectfully thank the authors for providing a high-quality and relevant rebuttal that strengthens their submission.
> >
> > In the following comments, I would like to continue the discussion with the authors regarding their responses.
> >
> > As a general comment, it would be helpful if you highlighted your changes in a different color in the revised manuscript.
> >
> >
> > ## Model training and architecture choices.
> >
> >
> > The provided arguments are relevant: canopy height estimation requires global context, in contrast to pixel-based or small patch-based approaches; the corpus size is substantial enough to train a large model from scratch; and the method does not require computational optimization tricks.
> >
> > As the authors mention, Swin Transformers include hierarchical feature learning, which has proven useful for dense prediction tasks. However, hierarchical patch structures have not been adopted in remote sensing foundation models for practical reasons: they are incompatible with current MAE or I-JEPA frameworks, which most large-scale models rely on. Highlighting this design choice—emphasizing the advantages of the Swin-based approach while acknowledging why current large-scale models have not adopted it—could strengthen the motivation in the related work section, as suggested in "Framing our contribution."
> >
> > That said, Tolan et al. pretrained a ViT using DinoV2 and fine-tuned it on GEDI pseudo ground truth labels. Although they do not achieve the best performance according to Table 1, their work demonstrates that adapting pretrained backbones with self-supervised learning (SSL) shows promise for large-scale canopy height estimation. Regarding the arguments in "Why train from scratch?", I would like to mention recent work from Siméoni et al. [1], which aligns with recent remote sensing literature: “In particular, SSL produces rich, high-quality visual features that are not biased toward any specific supervision or task, thereby providing a versatile foundation for a wide range of downstream applications.”
> > Additionally, Siméoni et al. [1] proposed a multi-stage SSL training approach, now more common due to increasing corpus sizes, and fine-tuned their backbone on canopy height mapping, outperforming Tolan et al. (see Section 8).
> >
> > Given the recent ArXiv publication of DinoV3 [1], I do not expect quantitative comparisons with their method. However, their approach shows promise despite your arguments, particularly given your statement that "training is generally not the main bottleneck."
> >
> > To what extent do you believe DinoV3 [1] and SSL pretraining methods in general are unsuitable for canopy height estimation, despite their use in related work?
> >
> > Would multi-stage pretraining leveraging the temporal information in your large-scale corpus be a promising direction for future work?
> >
> > [1] Siméoni et al., DinoV3. In ArXiv 2025.

---

> > > ### Comment · Reviewer_c1mT · 2025-11-24
> > >
> > > ## Architecture & Loss Ablation/Interplay
> > >
> > > The ablation study is appreciated for a rigorous study. Table 3 shows that there is no significant gain of using the GrowthLoss and the finetuning approach without the temporal information, which makes sense.
> > >
> > > Figure 8 shows that fine tuning with the GrowthLoss might lead to better time series predictions (purple cross), but not always since it does not detect the small drop between 2021 and 2022 for the green cross compared to the “pretrained model”.
> > > Note that without ground truth, it is also difficult to evaluate which model is actually correct in terms of predicted heights.
> > >
> > > Could it be feasible to define a GEDI time series at several locations (at least 3 points) to properly quantify the potential gain of the GrowthLoss?
> > >
> > > ## Metrics & Improvements
> > >
> > > I thank the authors for the details about the evaluation protocol and for the new Table 5, which shows strong performance for heights below 5 m, although GEDI labels may lack sufficient precision.
> > >
> > > However, it remains unclear why these predictions do not appear in Figure 4 (right), while all competing methods include a dedicated bin for predictions below 5 m. Would it be more appropriate to include these predictions in the plot and mention the threshold afterward? Otherwise, the visual comparison with other methods' predictions appears unfair.
> > >
> > > Considering the interquartile range (IQR) in Tables 1 and 5, should we interpret the proposed method as providing more stable results compared to competing methods? An additional interpretation would strengthen the modest gains in average performance.
> > >
> > >
> > >
> > > ## Train/Test Split
> > >
> > > My greatest concern is that spatial cross-validation, or at minimum train-test spatial splits, were not employed.
> > > Although a 360 m minimum distance threshold was established, there is no explanation of how this threshold was determined or whether it effectively removes spatial correlation.
> > >
> > > Spatial correlation is a well-known problem in machine learning applied to remote sensing that diverges depending on the downstream tasks, and spatial splits are a common best practice [2]. One way to verify that the authors' methodology is correct would be to compute a variogram that plots sample variance as a function of spatial distance to determine the range of spatial autocorrelation. Alternatively, any other statistical method demonstrating minimal spatial autocorrelation would be appreciated [2].
> > >
> > > [2] Salazar et al., Fair train-test split in machine learning: Mitigating spatial autocorrelation for improved prediction accuracy. In Journal of Petroleum Science and Engineering 2022.
> > >
> > >
> > > ## Height Errors and Above-Ground Biomass (AGB) propagation
> > >
> > > I thank the authors for this additional detail. Estimating high error propagation to AGB may be beyond the scope of this work due to the variability of coefficients required for local calibrations.
> > >
> > > ## Additional paragraphs and figures
> > >
> > > I thank the authors for all the provided additional information, including the limitations and future work sections.
> > > Figure 9 is quite interesting: it appears that average model performance increases over time while variability (IQR) decreases across all heights except 40–45 m, potentially suggesting that the model effectively leverages the time series to improve predictions. Does this interpretation align with your findings? Are there other conclusions from this plot that support the innovative approach of incorporating time series data for canopy height estimation?

---

> > > > ### Author Response · Authors · 2025-11-27
> > > >
> > > > We updated the paper to reflect all changes in blue color. For new or changed figures and tables, only the caption is in blue.
> > > >
> > > > ## Self-Supervised Learning
> > > > 1. **Updates to the Manuscript**: We have updated Section 2.1 to explicitly detail why we chose a hierarchical Swin backbone over isotropic Foundation Models. We clarify that Swin’s shifted-window attention is structurally incompatible with the random masking and crop-based invariance used in standard SSL frameworks (MAE, I-JEPA), despite Swin's superiority for dense prediction.
> > > >
> > > > 2. **Suitability of SSL (DINOv3)**: We fully recognize the immense value of SSL and agree that backbones like DINOv3 yield high-quality, robust features. However, a key limitation remains: current Foundation Models are predominantly spatial. Since our primary objective is spatiotemporal monitoring (ensuring consistency over 7 years), adapting a static spatial ViT to handle 4D temporal data is structurally complex and often suboptimal. Given that we are not data-constrained—possessing a massive labeled corpus—training a native spatiotemporal architecture from scratch proved more effective for modeling dynamics than retrofitting a spatial backbone.
> > > >
> > > > 3. **Future Work**: We agree that multi-stage pre-training leveraging temporal dynamics is the next logical step. We have added a reference to this in the discussion, citing recent work like OlmoEarth [2], SkySense++ [3] or AlphaEarth [4] as a promising direction for temporal Earth Observation.
> > > >
> > > > **References**:
> > > >
> > > > [2] Herzog et al., "OlmoEarth: Stable Latent Image Modeling for Multimodal Earth Observation," arXiv 2025.
> > > >
> > > > [3] Wu, K., Zhang, Y., Ru, L. et al. A semantic-enhanced multi-modal remote sensing foundation model for Earth observation. Nat Mach Intell 7, 1235–1249 (2025). https://doi.org/10.1038/s42256-025-01078-8
> > > >
> > > > [4] Brown, Christopher F., et al. "Alphaearth foundations: An embedding field model for accurate and efficient global mapping from sparse label data." arXiv preprint arXiv:2507.22291 (2025).
> > > >
> > > >
> > > > ## GEDI Time Series
> > > > We agree with the reviewer’s assessment of Figure 8. Without dense temporal ground truth, interpreting specific discrepancies involves some degree of hypothesis. While it is possible that the GrowthLoss smoothed over a real disturbance in that specific instance, our broader qualitative analysis suggests that the baseline (pretrained) model often exhibits spatial instability at forest borders—falsely depressing height values in pixels adjacent to disturbances. The GrowthLoss effectively mitigates this artifact, though we acknowledge the trade-off regarding potential over-smoothing of minor real-world drops.
> > > >
> > > > Regarding GEDI time series validation: This is unfortunately infeasible. GEDI is a sparse sampling mission, and its orbital tracks rarely repeat the exact same location. Even in rare cases of overlap, the combination of geolocation uncertainty (∼10 m), large footprint size (∼25 m), and inherent label noise obscures the subtle inter-annual height changes we aim to detect, preventing the extraction of a robust temporal signal.

---

> > > > > ### Author Response · Authors · 2025-11-27
> > > > >
> > > > > ## Figure 4 and IQR Interpretation
> > > > > We are sorry for the confusion. Figure 4 does not use any filters. As GEDI measures grass and bare ground at ~2.5m our models lowest prediction is 2.5m, which is reflected in both the scatterplot on the left, as well as the histogram on the right. All other maps either set some areas manually to 0m (using some forest mask, etc.) or train with ALS that correctly measures ground elevation at 0m.
> > > > >
> > > > > ## Spatial Autocorrelation
> > > > > We have added a new analysis (Figure 8) to the paper and clarify our experimental setup as follows:
> > > > >
> > > > > 1. **Statistical Verification (New Figure 8)**: To empirically verify our split, we computed a correlogram (to be more robust to outliers) (Pearson correlation coefficient of residuals vs. distance) as requested. As shown in the new Figure 8, spatial autocorrelation decays linearly from ∼0.75 at 0 m to a minimum of ∼0.52 at 400 m, after which it stabilizes/oscillates around 0.55. Although interpretation can only be done with caution, the peak at 600m coincidentally matches GEDI's intra-beam distance (each of the 8 GEDI beams is seperated by exactly 600m).
> > > > >
> > > > > 2. **Justification of the 360 m Threshold**: Our model consumes 10 m resolution patches with significant spatial context. To prevent any single pixel from appearing in both the "train" and "test" receptive fields, a buffer for each patch is required. The 360 m distance corresponds to the maximum extent of these buffered input tensors. The correlogram confirms that this design choice, though coincidental, aligns with the natural spatial correlation decay of the target variable, offering an additional justification for the threshold.
> > > > >
> > > > > 3. **Transductive vs. Inductive Setting**: Regarding the suggestion for strict spatial blocking (e.g., leaving out large contiguous regions): We respectfully argue that for our specific application — global-scale mapping — a transductive "gap-filling" approach is more appropriate than a strict inductive "extrapolation" approach.  Our goal is not to train a model that generalizes to unseen planets or biomes (extrapolation), but to upscale sparse GEDI samples to dense maps within existing biomes (interpolation). It is somewhat similar to Salazar et al. [2] Demonstration 3, with the difference that our Real-World (RW) use case is every pixel in the entire image. We do however acknowledge that this only works where GEDI data is available (unlike above 52N) and extrapolation is needed when going to Out-of-GEDI regions.
> > > > >
> > > > > ## Impact of Time on Model Performance
> > > > > While Figure 9 suggests a trend of improving model performance over time, this result must be interpreted with caution. Theoretically, this improvement is consistent with the model effectively leveraging accumulated historical context—analogous to a clinician improving their diagnosis as a patient's history becomes more complete. However, because our validation relies exclusively on GEDI labels, we cannot definitively disentangle true model gains from potential data artifacts. Confounding factors may include GEDI sensor degradation over time, shifts in label distribution due to varying cloud cover in complex terrains, or changes in orbital sampling patterns that might bias later years toward spatially 'easier' landscapes.
> > > > >
> > > > > We hope to have addressed your concerns. If there are remaining uncertainties, we are happy to discuss them.

---

### Official Review · Reviewer_G5Vh · 2025-10-27

**Soundness:** 3
**Presentation:** 3
**Contribution:** 3
**Rating:** 6
**Confidence:** 4

**Summary:**

SUMMARY: The paper focuses on the task of temporal tree canopy height estimation from remotely sensed data. As the authors outline, this task is relevant for global forest monitoring and directly impacts downstream applications like carbon stock or sequestration estimates. The authors aim to provide a global map - and the first map over time, allowing users to assess changes. Methodologically, the authors propose a new neural net architecture to model tree height, the Temporal-Swin-Unet. This is a combination of two existing methods, the Video Swin Transformer and the Swin Unet. The authors then evaluate their approach by comparing it to existing global tree height maps, showing consistently improved accuracy.

**Strengths:**

STRENGTHS:
I particularly enjoyed the following aspects of the paper:
- The paper is well motivated and tackles an incredibly important real-world task that is clearly suited for ML/vision models.
- The paper is very well written; it is easy to understand and has a good / intuitive flow.
- Part of that is the papers simplicity; the paper (mostly, exceptions below) has a great balance of depth and simplicity; the proposed methodological advancement is simple but quite elegant and well motivated by the problem setting.

**Weaknesses:**

SHORTCOMINGS:

I have two major concerns with the current draft of the manuscript:

- Part of the "growth loss" is the disturbance indicator. The paragraph introducing it is too short and it is unclear how the disturbance indicator motivated? choice of thresholds here seem arbitrary? The authors say that "A disturbance is considered to occur in zref ∈ RY when a) tree height decreased by more than 50% and more than 4 m and b) tree height decreased to less than 10 m within two years." (line 224), but the choice for these numbers are not explained at all and seem arbitrary. I assume there is some sort of expert knowledge behind them but this NEEDS to be explained!

- Crucial experiments on the performance over space and time are missing. It would be very important to know if the method performs equally well everywhere on the planet, or whether there are areas of higher and lower performance. This should follow e.g. the analysis in [1] (see Extended Data Fig. 1 in [1]). This sort of knowledge is very important for on-the-ground practitioners. Secondly, an analysis of the performance over time would be equally interesting / important. I see that the authors say that "Due to the sparse temporal and spatial distribution of GEDI labels, a temporal validation with GEDI is not possible. " (line 366) - What is meant by this exactly? Are GEDI labels not spatio-temporally aligned (e.g. a given location only occurs on one time step)? You should still be able to assess temporal performance by averaging errors for a given time step. Am I missing something?

- A more minor point is that I would like to see some discussion of the applicability of the method to local height mapping problems. Specifically [2] argues that these sort of global tree height maps fall short of being actually useful in many local applications. I would be curious how the authors contextualize their work within this critique.

**Questions:**

Overall this a paper tackling a relevant real-world problem and introducing an intuitive new method. I'd ask the authors to consider my questions and concerns in the "weaknesses" section.

References:

[1] Lang, Nico, et al. "A high-resolution canopy height model of the Earth." Nature Ecology & Evolution 7.11 (2023): 1778-1789.

[2] Rolf, Esther, et al. "Contrasting local and global modeling with machine learning and satellite data: A case study estimating tree canopy height in African savannas." arXiv preprint arXiv:2411.14354 (2024).

---

> ### Author Response · Authors · 2025-11-21
> **Rebuttal**
>
> We thank the reviewer for the positive feedback. Below, we address the raised points in detail:
>
> ### Growth-Loss/Disturbance-Indicator
> That is totally right. Both the specific values and the overall structure of the loss were developed in close collaboration with forest-ecology experts, and we have revised the manuscript to better reflect this point. However, these discussions also revealed substantial uncertainty, with differing expert opinions on the most appropriate formulation. Since no suitable empirical dataset exists to derive or validate such a structure directly, we necessarily rely on expert knowledge for defining these constraints.
>
> ### Time/Space-Experiments
> We added a figure illustrating the spatial distribution of errors (Figure 11), following the style of Lang et al. However, we would like to emphasize the inherent limitations of such an analysis. As shown in Figure 5 (previously Figure 6) and noted throughout the literature, the prediction error is strongly correlated with tree height. As a result, global error maps often largely reflect the underlying distribution of tall versus short canopies. To mitigate this effect, we generated separate maps for six tree-height intervals (0-30 m in 5 m bins). This stratification reduces the height-error coupling and highlights more meaningful regional patterns, for example, comparatively lower errors for tall canopies in the Amazon and Congo Basin relative to other regions.
>
> ### Necesity of global-scale tree height products
> We acknowledge the limitations of global tree-height products in light of this discussion. When high-quality local datasets are available, they should indeed be preferred, and global models should ideally be designed so that fine-tuning can further improve accuracy.
>
> Nonetheless, a few important considerations remain:
> 1. Fine-tuning a local model requires suitable local reference data, which is often unavailable.
> 2. While some forest research groups have the expertise and resources to train or adapt models, many others (like small forest owners for example) do not, or prefer not to engage in model development and instead rely on ready-to-use products.
>
> We hope we have addressed your concerns. If further clarification is needed on any point, please let us know.

---

### Official Review · Reviewer_P5pZ · 2025-11-01

**Soundness:** 4
**Presentation:** 4
**Contribution:** 3
**Rating:** 6
**Confidence:** 4

**Summary:**

This work introduces *ECHOSAT*, a global 10m tree canopy height time series spanning 2018–2023. A Vision Transformer performs pixel-level temporal regression on multi-year satellite imagery (Sentinel-2, Landsat) and sparse GEDI height labels. A novel growth loss is introduced to regularize predictions to follow realistic tree dynamics (gradual growth, sudden loss from fires/logging) without need for post-processing. The model improves over prior single-year SOTA methods on held-out GEDI data (RMSE=10.87m) and demonstrates disturbance detection (F1=0.82). Public release of height maps is planned.

**Strengths:**

- **Originality**: The growth loss enforces monotonic height increase and abrupt drops; this is the first global 10m multi-year canopy height map with inherent temporal modeling
%%previous work looks at single years only.

- **Quality**: The model uses multi-sensor data and sparse GEDI labels for GT. Ablations isolate growth loss impact on held-out GEDI (Table 1, p. 8).

- **Clarity**: The paper is very well-written; it's clearly structured, with well-explained methods; outlines explicit contributions, provides clear mathematical formulations, and effective visuals.
- **Significance**: The resulting height map time series supports global-scale monitoring of forest growth and disturbance, with applications in carbon accounting and climate mitigation. The planned public release of height maps is a valuable contribution.

**Weaknesses:**

- **Clarity**: Equations are unnumbered, making referencing difficult.
- **Significance**: Height-to-carbon flux not evaluated -- above-ground biomass (AGB) to CO₂ conversion or flux tower validation would enhance climate impact, i.e. in carbon accounting.

- **Originality**: The main novelty lies in the growth loss (Sec 3.3), but a comparison to learned temporal dynamics in *TimeSformer* (Bertasius et al., ICCV 2021) or *EarthFormer* (Gao et al., NeurIPS 2022) would help quantify value the loss adds beyond attention-based modeling.
- **Quality**: Results are strong against modern single-year baselines, but temporal SOTA comparisons could further strengthen the authors' claims.

**Questions:**

1. Consider comparing to *TimeSformer* or *EarthFormer* -- Replacing your ViT encoder with one of these approaches that learn temporal dynamics via space-time attention would help clarify the advantage your growth loss provides.

2. Validate carbon flux using height -- Height is a solid proxy, but for carbon accounting claims, it's valuable to estimate flux by converting your maps to AGB then CO₂  uptake/release using allometry like Jucker et al. (2022), validated against a flux tower site (e.g., Harvard Forest with public data).This additional step would ground the significance in real carbon metrics and strengthen the climate impact from my perspective.

3. To improve mathematical clarity, consider numbering equations -- this would make it easier to reference formulations like the growth loss and follow derivations.

---

> ### Author Response · Authors · 2025-11-21
> **Rebuttal**
>
> We thank the reviewer for the feedback on our paper. We will address the concerns raised in detail below:
>
> ### Clarification of contribution
> We would like to clarify that ECHOSAT covers the years 2018 till 2024 and is only based on Sentinel-2 imagery for the optical modality, not on Landsat data. Furthermore, there appears to be some confusion or misattribution regarding the disturbance-detection literature: while the disturbance indicator could be used to detect canopy disturbances from tree height, this application is neither explored nor evaluated with F1 scores in our work. While Table 1 only reports metrics comparing the existing methods, we have added Table 3 to provide an ablation on the impact of our GrowthLoss compared to an L1 and a simple linear regression.
>
> ### Significance
> We agree that linking tree height to carbon flux is highly relevant and represents an important downstream use of the ECHOSAT maps. Demonstrating how height predictions and their uncertainties translate into above-ground biomass estimates would indeed strengthen confidence for carbon accounting. In our case, however, this lies beyond the scope of the paper. Our focus is on producing the ECHOSAT maps and documenting the underlying data sources, preprocessing steps, model design, loss formulation, and direct height-prediction evaluation. To provide an additional point of reference, we included a comparison with LiDAR-derived labels from the Lidar HD Campaign over the Landes Forest (see Figure 10). Because LiDAR data are only sparsely available, the scatterplot reflects performance solely for vegetation conditions in the Landes Forest. These results should therefore not be interpreted as representative of other regions.
>
> ### Comparison to existing architectures
> To select an appropriate encoder backbone, we reviewed existing architectures and determined that, while not originally developed for remote sensing, the Video-Swin-Unet was the most suitable for our application. Our reasoning was threefold:
>
> 1. Earlier models such as EarthFormer and TimeSformer (we assume you mean the ICML 2021 one) are relatively dated, with more recent architectures offering improved performance.
> 2. Foundation models (e.g., Galileo, CROMA) would require substantial adaptation to handle our specific task, which diminishes the benefit of pretraining.
> 3. To our knowledge, existing foundation models do not employ hierarchical attention, which has been shown to enhance performance in image segmentation and pixel-level regression.
>
> We subsequently explored adaptations to efficiently integrate multiple data sources and to compress the temporal dimension from twelve tokens per year into a single token per year.
>
> ### Clarity
> Thank you for pointing this out. We have added numbers to the important equations.
>
> If any points remain unclear, we welcome further questions and are happy to provide a more in-depth discussion on specific aspects of the work.

---

### Official Review · Reviewer_kAKD · 2025-11-03

**Soundness:** 3
**Presentation:** 4
**Contribution:** 2
**Rating:** 4
**Confidence:** 5

**Summary:**

ECHOSAT provides a 50TB multi-modal (radar, multi-spectral, LiDAR), multi-temporal (monthly composites from 2018 to 2024) dataset from Earth observation with 3 million globally sampled geolocations of shape 18x84x96x96 (= channel x time x widht x height) at 10m pixel resolution. The authors explore the performance (Tab. 1, Figs. 3 & 4) of a Video-Swin-UNet-like architecture (Fig. 1) to predict tree height as determined by the GEDI sensor (mounted onto the ISS) when the network is trained by an additional loss that restricts tree growth to physical bounds (Sect. 3.3). Results indicate an advantage over existing methods.

**Strengths:**

The paper is clearly structured, written in plain English, with the main text accommodated by illustrative figures, equations, tables, and appendices with additional details. The dataset is carefully curated (App. A.2.1) and the methodology well documented (App A.2.2). Experimental results are cleanly evaluated against existing methods (Sect. 4.3).

**Weaknesses:**

The work falls short in major novelties for the ICLR community regarding learning representation methods. While ECHOSAT resembles a valuable dataset for the Earth observation community, the Temporal-Swin-Unet (Sect. 3.2) blends minor adjustments (1x1 patch size, additional layers and skip connections) from existing architecture. The additional *Growth Loss* (Sect. 3.3) is specific to the application of tree height mapping, and resembles a neat, but limited innovation.

Unfortuantely, the authors evaluate the model performance on (hold-out) GEDI data the model was trained on. An independent modality to verify the temporal evolution of tree heights predicted, and a qualitative comparison to corresponding field surveys is missing. However, I appreciate the author's discussion of qualitative investigations such as in Fig. 7.

Given the Earth observation modalities fused ship in various spatial resolutions, I would appreciate a more detailed discussion around upsampling strategies to 10m per pixel, and their consequences. In particular, the label source GEDI for tree height estimation probes geospatial scenes at about 25 meter footprints. Discontinuities in tree height are common at forest boundaries and in areas disturbed by wild fires and logging.

I rate the paper a valuable scientific piece of work carefully conducted in general, but I believe it would better fit the scope of an Earth observation conference, a computer vision conference with geospatial tracks, or a high-profile domain journal. However, if other reviewers read the paper and consider my input to come to the conclusion this work fits the scope of ICLR, I am fine with acceptance.

- typos:
  * l105: typo _captures_ to _capture_
  * l633: $20\circ$ to $20^\circ$

**Questions:**

- Which license will the ECHOSAT dataset and the Temporal-Swin-Unet be published under?
- Please provide a table with dataset and its source utilized. In particular which datasets have been pulled from Google Earth Engine, and how was geospatial alignment implemented?

---

> ### Author Response · Authors · 2025-11-21
> **Rebuttal**
>
> We thank the reviewer for the feedback. To our understanding, you acknowledge the effort invested in ECHOSAT, but raise questions about performance evaluation, contribution significance, and data preparation. We have tried to address these concerns below. For a full list of changes made to our paper, please see the official comment.
>
> ### Clarification on Paper Goal
> In our paper we describe the dataset in detail, with all the used sources, applied filtering and final sizes. However, we would like to point out that we see our main contributions elsewhere (refer to "Contributions" in Section 1): ECHOSAT is the map itself, a high-resolution (10 m), temporal (annually 2018-2024) and global tree height map. This map provides the first globally consistent tree-height product with annual updates at 10 m resolution, offering a practical basis for investigating year-to-year canopy changes and related ecosystem processes.
>
> ### Validation with High-Quality Labels
> Validating large-scale tree-height products remains challenging. While GEDI offers global coverage, its measurements are known to have notable noise and uncertainty. Airborne LiDAR (ALS) provides more accurate reference data, but such datasets are sparse, geographically biased toward economically well-surveyed regions, and, to date, we have not identified ALS sources with suitable temporal revisits for year-to-year evaluation. Given these limitations, we consider GEDI a reasonable benchmark: although not perfect, it is globally distributed and consistently available. In addition, we performed an evaluation with ALS-derived label in the well-studied Landes forest in France and included the results in the Appendix (see Figure 10).
>
> ### Suitability for ICLR
> Although the individual techniques used (e.g., model architecture, loss formulation, data preprocessing) are not themselves novel, a global application of this scale benefits from a solution that is simple, reliable, and effective. Because the model must be applied to 1.5 PB of data, we need to balance accuracy, efficiency, and data availability. We are confident that the chosen approach provides a strong compromise across these dimensions.
>
> ### Data sources and publication
> Thank you for pointing out this issue. In the revised version of our manuscript, we have added an overview table in the appendix and expanded Section A.2.1 with additional information on spatial alignment. With respect to GEDI, it is worth noting that the commonly cited footprint diameter of 25 m only depicts an approximate value; in practice, we find that geolocation uncertainty has a much larger impact on label consistency than the footprint size itself (see Figure 7 for an illustration of the footprint radius).
>
> The ECHOSAT map and the training code will be publicly hosted on Google Earth Engine and GitHub, respectively, under a CC BY 4.0 license, enabling free use, modification, and redistribution. Because hosting a 50 TB dataset publicly is nontrivial, we will provide access to the full dataset upon request.
>
> Should you wish us to elaborate on any particular point, we would be happy to do so.

---

### Author Response · Authors · 2025-11-21
**Rebuttal**

We thank all reviewers for their helpful feedback. In response, we revised the manuscript and made several additions (changes have been marked in blue, for new changed figures only the caption is blue) and clarifications:

**Structural and Clarity Improvements**
- Added a dedicated Limitations section (Section 5.1).
- Added related work on self-supervised learning (Section 2.1).
- Included an overview table summarizing all data sources, their resolution, resampling and other information (Table 2).
- Improved clarity regarding the disturbance indicator (now explicitly noted as based on expert knowledge).
- Clarified details about the train/test split distance (Appendix 2.2).
- Corrected minor typos.

**New Figures and Visual Analyses**
- Moved the boxplots showing errors for different tree heights(Figure 5) from the appendix to the main paper.
- Added Figure 7 to illustrate GEDI’s 25 m footprint and photon density for different sized and positions trees.
- Added Figure 9 (and Table 3) to present the requested ablation study on fine-tuning (GrowthLoss vs. linear regression vs. none).
- Added Figure 8 to analyse spatial autocorrelation with increasing distance.
- Added a boxplot comparing errors for different years across height bins (Figure 10).
- Added a comparison to LiDAR with data from the French Landes Forest (Figure 12).
- Added a spatial error distribution map similar to Lang et al. (Figure 11).

**Expanded Quantitative Results**
- Added IQR to all metric tables.
- Added a new metrics table for labels < 5 m, analogous to Table 1 (Table 5).
- Added Table 3 for quantitative results from the ablation study (see above).

---

### Author Response · Authors · 2025-12-03

We appreciate the feedback provided by the reviewers. In our response, we have clarified our design choices regarding the model architecture and training paradigm, added an ablation study to substantiate the proposed GrowthLoss, and included new figures to deepen the analysis. Furthermore, we have addressed the concerns regarding spatial validation with additional empirical evidence. We specifically acknowledge the constructive exchange with Reviewer c1mT, which helped refine these aspects of our work.

---

### Meta-Review · Area_Chair_Y1Fp · 2026-01-04

**Summary:**

Several reviewers have concerns over the evaluation and analysis of the method, including the reliance on hold-out GEDI data for training and testing (kAKD), and the lack of performance analysis over space and time (G5Vh). Also, they mentioned a limited scope, the need for comparison to temporal SOTA methods like TimeSformer and EarthFormer (P5pZ), and for more ablation studies to justify the architecture and Growth Loss (c1mT). A reviewer also mentioned the limited novelty for the ICLR community (kAKD). Reviewers are also looking for better motivation and justification on the Growth Loss (G5Vh, P5pZ). Other minor issues include missing related work on spatio-temporal methods and remote sensing foundation models (c1mT), unnumbered equations (P5pZ), and an insufficient discussion of the applicability to local height mapping and the exclusion of labels below 5m (G5Vh, c1mT).

**Reviewer Concerns:**

While the minor issues were addressed in the rebuttal. The main concerns are not fully addressed.

For example, Reviewer kAKD's concern regarding relevance and novelty. The authors responded with “Although the individual techniques used (e.g., model architecture, loss formulation, data preprocessing) are not themselves novel, a global application of this scale benefits from a solution that is simple, reliable, and effective. Because the model must be applied to 1.5 PB of data, we need to balance accuracy, efficiency, and data availability. We are confident that the chosen approach provides a strong compromise across these dimensions.” From the AC’s perspective, it further emphasizes the point of the reviewer.

The authors also responded to the reviewer regarding comparisons with SOTA by explaining that they require “substantial adaptation” and are different in “not employ hierarchical attention”. The AC believes that the reviewer asked for these comparisons because they believe that such comparisons are a reasonable request. Hence, the response will not address the reviewers’ concern.

**Reviewer Scores:**

From the AC’s read of the rebuttal and review, the AC found that the reviewers would have kept their scores. With reviewer kAKD potentially lowering the score, as another reviewer echoed the concern. As the reviewers pointed out, the paper’s contribution is probably a better fit for another venue that focuses more on the application/science side of the work. The innovation regarding the learning of representation, and the over system would be quite limiting to the ICLR audience (at least to 2 out of the 4 reviewers).

---

### Decision · Program_Chairs · 2026-01-26

Reject